# Understanding Truncated Positional Encodings for Graph Neural Networks

James Flora [* 1]   Mitchell Black [* 2]   Weng-Keen Wong [1]   Amir Nayyeri [1]

## Abstract

Positional encodings (PEs) enhance the power of graph neural networks (GNNs), both theoretically and empirically. Two of the most popular families of PEs—spectral (e.g., Laplacian eigenspaces, effective resistance) and walk-based (polynomials of the adjacency matrix)— are theoretically equivalent in expressive power, with expressivity between the 1-WL and 3-WL tests. However, this equivalence assumes the GNN uses the "complete" version of these PEs, which requires $O(n^3)$ time and space complexity. Instead, practitioners commonly use truncated variants of these encodings, such as the first $k$ eigenspaces or powers of the adjacency matrix. However, the theoretical properties of these truncated PEs are unknown. In this work, we initiate the study of these truncated PEs. Theoretically, we show that, under truncation, several families of PEs are fundamentally different in expressive power. As a corollary, we show that truncated spectral PEs are no longer stronger than the 1-WL test. We also study a family of spectral PEs, the $k$-harmonic distances, to highlight the differences in expressive power of even closely related truncated PEs. Finally, we experimentally show that a mix of truncated PEs is preferable to any single family on real-world datasets.

## 1. Introduction

Graph Neural Networks (GNNs) have been extensively studied in the machine learning community. GNNs have demonstrated strong performance on a variety of tasks, including node classification, graph classification, and regression. However, GNNs are fundamentally limited in their ability to capture global structural information, which is essential for many graph learning problems.

This limitation has been formalized (Xu et al., 2018; Morris et al., 2019) by showing that message-passing neural networks (MPNNs), perhaps the dominant class of GNNs, are no more powerful than the Weisfeiler-Lehman (WL) test (Weisfeiler & Leman, 1968).

It is important to situate such expressivity results in the broader context of graph learning. In many real-world benchmarks, most graphs are distinguishable by the WL test (Zopf, 2022). That said, this does not mean that MPNNs can perfectly learn these benchmarks, as downstream tasks do not necessarily reduce to deciding graph isomorphism. Rather, expressivity results serve as a principled proxy for the *structural sensitivity* of a model: they characterize which structural properties of the graph can computed by the model. If two graphs differ in some property but are indistinguishable by a family of GNNs, then this family of GNN cannot compute this property. In practice, these properties do not simply determine whether two graphs are distinguishable, but may instead correspond to task-relevant signal such as centrality, motif counts, or long-range dependencies that correlate with the prediction task.

As an alternative to MPNNs, graph transformers (GTs) have recently been proposed. GTs have shown potential to both increase the expressive power of GNNs beyond MPNNs and address issues like oversmoothing (Oono & Suzuki, 2020; Cai & Wang, 2020) and oversquashing (Alon & Yahav, 2021). However, GTs rely on the use of *positional encodings* (PEs) to input the graph to the network. PEs can provide the missing global structural information in the form of node or edge features, and can curb the issue of oversmoothing by requiring fewer layers of message passing or attention for this global structural information to propagate through the graph. However, like MPNNs, PEs also have bounded expressivity, and no known polynomial-time-computable and polynomially-sized PE captures all information about the graph.

Two of the most popular families of PEs are spectral (Lim et al., 2023; Huang et al., 2024; Zhang et al., 2024; 2023) and walk-based encodings (Ma et al., 2023; Gai et al., 2025). Spectral PEs are derived from the eigenvalues and eigenvectors of the graph Laplacian. Walk PEs concate-

---

[*]Equal contribution  [1]School of Electrical Engineering and Computer Science, Oregon State University, Corvallis, Oregon, USA [2]Halıcıoğlu Data Science Institute, University of California San Diego, San Diego, California, USA. Correspondence to: James Flora <floraj@oregonstate.edu>.

*Proceedings of the $43^{rd}$ International Conference on Machine Learning*, Seoul, South Korea. PMLR 306, 2026. Copyright 2026 by the author(s).

nate powers of the adjacency matrix. Spectral and walk PEs are known to have equivalent expressive power when $\Omega(n)$ powers of the adjacency matrix are used (Black et al., 2024b; Gai et al., 2025), and both are known to be lie between the 1-WL and 3-WL (2-FWL) tests in the WL expressivity hierarchy.

While spectral and walk positional encodings are equivalent to one another in theory, these PEs require a sizable amount of computation for most real world tasks, requiring a $O(n^3)$ time pre-processing step for every graph in the training data, and $O(n^3)$ time and space complexity for graph transformers. As such, practitioners often use truncated versions of these encodings (e.g., using the first $k$ eigenvectors/eigenspaces of the Laplacian or first $k$ powers of the adjacency matrix) to see some of the benefit from using the encoding while avoiding the prohibitive overhead cost. However, how these truncated representations compare in practice has not been studied.

### 1.1. Our Contributions

We aim to fill this gap in the research by comparing fixed-size spectral and walk PEs both theoretically and empirically. First, we show that truncated spectral and walk encodings have very different expressive powers. Specifically, we show that constant-sized versions of one can distinguish graphs that size $\Omega(n)$ versions of the other cannot. Moreover, we show the surprising result that $\Omega(n)$-size spectral encodings can be less expressive than 1-WL test. In short, while spectral and polynomial positional encodings are equivalent in their complete form, they carry different information about the graph in their truncated form.

Another common and well-studied positional encoding for graph transformers is the *resistance distance* (Zhang et al., 2023) (aka effective resistance.) We show that the effective resistance is a weaker PE than the eigenspace projections and walk PE on weighted graphs. Additionally, we show that the effective resistance is not stronger than the weighted analog of the 1-WL test, so GTs using effective resistance are not stronger than message-passing network that use the edge weights.

As the effective resistance alone is not as powerful as the eigenspace projection and walk PEs, we consider a family of spectral positional encodings called the $k$-harmonic distances (Black et al., 2024a) that generalize the resistance distance.[1] We argue that the $k$-harmonic distances are a natural bridge between spectral and polynomial positional encodings. First, we prove that using $\Theta(n)$ $k$-harmonic distances has equivalent expressive power to using the complete spectral or walk positional encodings, showing that spectral and walk encodings are only two of potentially

---
[1]The squared 1-harmonic distance is the effective resistance.

many positional encodings with the same expressive power, opening the door for alternative positional encodings (Theorem 4.6). We show that even within the family of $k$-harmonic distances, fixed-size encodings can have different expressive power. For MPNNs, the 2-harmonic (biharmonic) distance can distinguish pairs of graphs in a constant number of layers while the resistance distance needs $O(n)$ layers. While this theorem is less general than previous results, it illustrates the trade-offs between positional encodings in the same family. Conversely, we prove that if some $k$-harmonic distance is able to distinguish a pair of graphs, then most other $k$-harmonic distances can distinguish these graphs as well.

As we show the different truncated PEs can have very different expresssive power, we propose the following rule-of-thumb: if you are going to use truncated PEs, mix PEs from different families. We confirm this principle on real-world benchmarks, where we show that using a mix PEs from different families achieves superior performance than using PEs from a single family.

## 2. Related Work

**Positional Encodings in GNNs**  While we consider a fine-grained approach in using truncated positional encodings, there have been many previously proposed methods and theory for the full versions of these positional encodings. The first graph transformers used Laplacian eigenvectors as positional encodings (Dwivedi & Bresson, 2021) with subsequent works following suit (Kreuzer et al., 2021; Rampasek et al., 2022; Zhou et al., 2024). However, Laplacian eigenvectors suffer from sign and basis ambiguities, so Lim et al. (2023) proposed to use the projection onto the eigenspaces, rather than the eigenvectors themselves, to avoid this ambiguity. Huang et al. (2024) and Zhang et al. (2024) proposed alternative techniques using the projections onto the eigenspaces, which we make use of as a type of "truncated" encoding.

**$k$-harmonic distances**  Though the $k$-harmonic distances are a natural extension of the effective resistance and represent a type of truncated spectral information, they were only recently proposed (Black et al., 2024a) and has not been fully explored. Its efficacy has been shown for clustering and as a centrality measure, but little else is known.

The effective resistance is the best studied $k$-harmonic distance in the literature, and it has countless applications across graph theory. For GNNs, effective resistance has been shown to be a useful positional encoding for MPNNs (Velingker et al., 2023) and GTs (Zhang et al., 2023). Further, it has been shown that MPNNs that make use of effective resistance are strictly more expressive than the WL test (Velingker et al., 2023).

The biharmonic distance has been used in the study of consensus networks (Yi et al., 2018b) and has been shown to be a measure of edge centrality (Yi et al., 2018a; Li & Zhang, 2018; Black et al., 2024a). However, it has not been studied as a positional encoding in GNNs.

**Expressivity of Positional Encodings**  Understanding the expressive power of PEs to distinguish non-isomorphic graphs has recently been an active area of research as it relates to graph learning, specifically in graph transformers (Zhang et al., 2023; 2024; Black et al., 2024b). Importantly, it is generally known that the expressivity of most positional encodings (including walks, effective resistance, shortest path distance, and eigenspace projections) lies between between the 1-WL test and the 3-WL tests.

## 3. Background

Let $G = (V, E)$ be an undirected, unweighted graph. Denote the number of vertices and edges as $n = |V|$ and $m = |E|$. Additionally, let the graph have a set of node features, $\{x_v \in \mathbb{R}^d : v \in V\}$, and a set of edge features, $\{e_{uv} \in \mathbb{R}^f : (u, v) \in E\}$. The **adjacency matrix** of $G$ is the matrix $A \in \mathbb{R}^{n \times n}$ where $A_{i,j} = 1$ if $(i, j) \in E$ and 0 otherwise. The **degree matrix** is the diagonal matrix $D \in \mathbb{R}^{n \times n}$ where $D_{i,i} = \deg(i)$. The **Laplacian matrix** is $L = D - A$ and is the central structure of study in spectral graph theory. L is positive semidefinite. The eigenvalues of the Laplacian are the **spectrum** of the graph. Throughout this paper, multisets are denoted with the double curly bracket notation $\{\!\!\{\,\}\!\!\}$.

**Graph Neural Networks**  *Graph neural networks (GNNs)* are functions that take as input a graph $G = (V, E)$, a set of node features, $\{x_v \in \mathbb{R}^d : v \in V\}$, and (optionally), a set of edge features $\{e_{uv} \in \mathbb{R}^f : (u, v) \in E\}$. The GNN iteratively updates the nodes features, where $h_v^{(t)}$ denotes the node features at layer $t$. Initially, $h_v^{(0)} = x_v$. We use the terminology graph neural network to encompass both message-passing neural networks and graph transformers.

**Message Passing Neural Networks**  The most common type of graph neural network are *message passing neural network (MPNNs)* (Gilmer et al., 2017). Each layer of an MPNN updates the feature of a node $h_v^{(t)}$ by aggregating the feature from its neighbors $h_u^{(t)}$ and (optionally) the features of the incident edges $e_{uv}$. For each layer $t \in \{1, ..., T\}$, a message passing layer updates the node features using the following formula:

$$h_v^{(t)} = \phi^{(t)}\left(h_v^{(t-1)}, \psi^{(t)}\left(\{\!\!\{(e_{uv}, h_u^{(t-1)}) : (u, v) \in E\}\!\!\}\right)\right),$$

where $\phi^{(t)}$ and $\psi^{(t)}$ are learnable functions and $\psi^{(t)}$ is invariant on multisets, e.g. mean, sum, max.

**Positional Encodings**  Positional encodings are functions defined on a graph that describe its topology. In this work, we are primarily concerned with relative positional encodings: encodings that associate values with each pair of nodes. A *relative positional encoding* (RPE) is a map $\psi$ that associates to any graph $G$ and map $\psi_G : V_G \times V_G \to \mathbb{R}^d$ such that for any two isomorphic graphs $G$ and $H$ and isomorphism $\sigma : V_G \to V_H$, it follows that $\psi_G = \sigma \circ \psi_H$.

**Graph Transformers**  More recently, *graph transformers* have been proposed as an alternative to message passing by replacing neighborhood aggregation with global self-attention over all nodes. While there have been may proposed ways of adapting the transformer for graphs, we use the *Graphormer-GD* architecture, as it is a strong way of incorporating RPEs into a transformer. In a Graphormer-GD layer, each node feature $h_v^{(t)}$ is updated by attending to the features of all nodes $u \in V$. Specifically, for each head $h \in \{1, ..., H\}$, queries, keys, and values are computed as linear projections $Q_v = h_v^{(t-1)}W_Q^h$, $K_u = h_u^{(t-1)}W_K^{(h)}$, $V_u = h_u^{(t-1)}W_V^{(h)}$, and attention weights are given by

$$A_{vu}^h = f_1(\psi_G(u, v))\text{softmax}_u\left(\frac{Q_v K_u^T}{\sqrt{d_h}} + f_2(\psi_G(v, u))\right)$$

where $f_1$ and $f_2$ are two functions (typically MLPs) and $\psi_G(v, u)$ denotes some positional encoding between nodes $v$ and $u$. The node features are then updated as

$$h_v^{(t)} = h_v^{(t-1)} + \sum_{h=1}^{H} \sum_{u \in V} A_{vu}^{(h)} V_u W_O^{(h)}$$

followed by a node-wise feed-forward transformation.

**WL Tests**  The *Weisfeiler-Lehman (1-WL) test* is an iterative algorithm that assigns labels to nodes in order to deduce whether or not two graphs are isomorphic. Specifically, the 1-WL test assigns each vertex $v \in V$ a color $\chi^{(t)}(v)$ for all $t \geq 0$. The labels are initialized to some arbitrary constant in the $0^{\text{th}}$ iteration, e.g., $\chi^{(0)}(v) = 1$ for all $v \in V$. For $t \geq 1$, the 1-WL color of a vertex $v$ is defined

$$\chi^{(t)}(v) = \text{hash}\left(\chi^{(t-1)}(v), \{\!\!\{\chi^{(t)}(u) : (u, v) \in E\}\!\!\}\right)$$

where $\text{hash}$ is an injective hash function.

Two graphs $G$ and $H$ are **indistinguishable** by the WL test if they have the multisets of colors for all $t \geq 0$, or formally,

$$\{\!\!\{\chi^{(t)}(v) : v \in V_G\}\!\!\} = \{\!\!\{\chi^{(t)}(v) : v \in V_H\}\!\!\}. \quad (\forall t \geq 0)$$

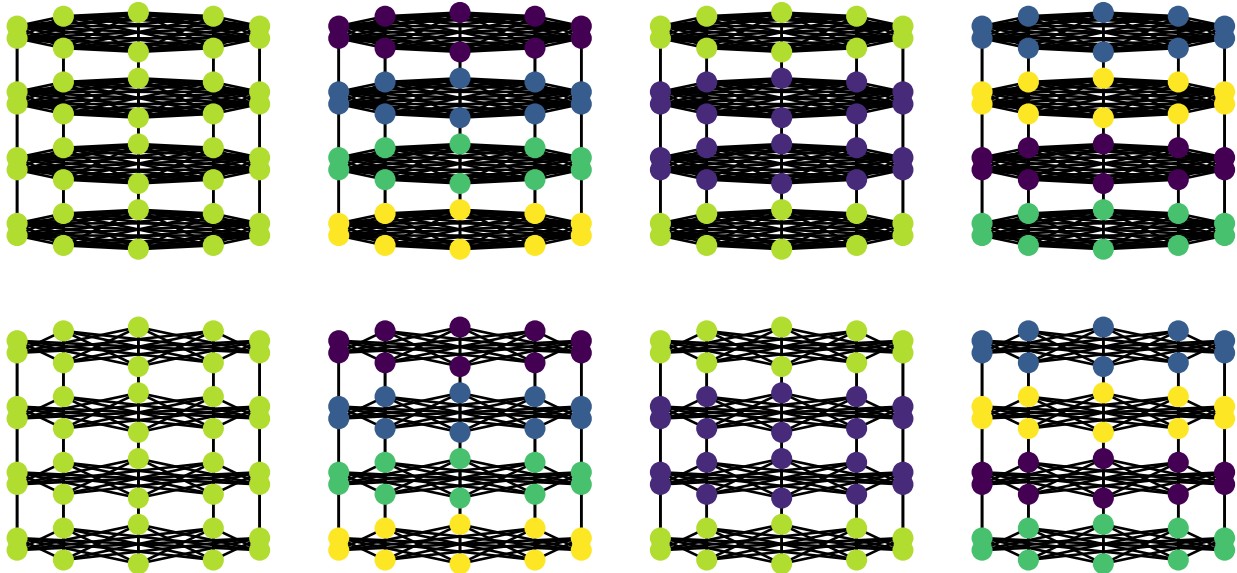

*Figure 1.* Top row: The four smallest eigenvectors of the graph $P_4 \times K_{10}$. Bottom row: The four smallest eigenvectors of the graph $P_4 \times K_{5,5}$. Comparing the two rows shows there is a bijection between the nodes of $P_4 \times K_{10}$ and $P_4 \times K_{5,5}$ that preserves the values of the eigenvectors. These two graphs are indistinguishable by 4-EP-WL.

The 1-WL test and MPNNs are known to be equal in their power to distinguish non-isomorphic graphs (Xu et al., 2018; Morris et al., 2019).

**Theorem 3.1** ((Xu et al., 2018; Morris et al., 2019)). *Two graphs are indistinguishable the WL test iff they are indistinguishable by all MPNNs.*

Further, there are higher order variants of the 1-WL test called the ***k-WL tests*** that assign colors to tuples of $k$ nodes rather than to single nodes; see (Huang & Villar, 2021).

**GD-WL** The 1-WL provides a bound on the expressive power of an MPNN. We would like to have similar tests that provide an upper bound on other types of GNNs, especially those that use PEs. Given an RPE $\psi$, we can characterize the power of a transformer using $\psi$ as a positional encoding using a variant of the WL test. The ***GD-WL test with RPE $\psi$*** (Zhang et al., 2023) (or ***$\psi$-WL*** for short) assigns a color to each vertex $v \in V_G$ according to the rule

$$\chi_\psi^{(t)}(v) = 0 \qquad\qquad\qquad (t=0)$$
$$\chi_\psi^{(t)}(v) = \left\{\!\!\left\{ \left(\chi_\psi^{(t-1)}(u),\, \psi(u,v)\right) \,:\, u \in V_G \right\}\!\!\right\} \quad (t \geq 1)$$

When Graphormer-GD uses $\psi$ as an RPE, it is equally powerful at distinguishing graphs as the $\psi$-WL test.[2]

---

[2]Stoll et al. (2026, Theorem 1) recently proved that a simplified version of Graphormer-GD that does not use the functions $f_1$

**Theorem 3.2** ((Zhang et al., 2023, Theorem E.3)). *Let $\psi$ be an RPE. Then two graphs are indistinguishable by $\psi$-WL iff they are indistinguishable by all Graphormer-GDs using $\psi$ as an RPE.*

# 4. Truncated Positional Encodings

In this section, we prove several properties of truncated positional encodings, including truncations of the eigenspace projections (Lim et al., 2023; Huang et al., 2024; Zhang et al., 2024), powers of the adjacency matrix (Ma et al., 2023), and the effective resistance (Zhang et al., 2023). We generally show these truncated PEs have very different expressive power. In doing so, we are able to show that it is possible for these encodings to be weaker than natural extensions of the 1-WL test, generally making them incomparable to the WL hierarchy.

## 4.1. Eigenspace Projections

The eigenvectors and eigenvalues of the graph Laplacian are powerful descriptors of a graph, encoding information like cluster structure (Lee et al., 2014). The original graph transformers using the Laplacian eigenvectors used them in their vector form, either adding them or concatenating them to the node features. However, this way of using

---

has equivalent expressive power to the original Graphormer-GD. Our experimental work was started before the publication of this paper so uses the older Graphormer-GD architecture.

Laplacian eigenvectors suffered from the fact that eigenvectors are not unique (they have sign and basis amiguity). Later, Lim et al. (2023) observed that the eigenvectors could be uniquely described by the projection matrices onto the eigenspaces. Zhang et al. (2024) summarized the spectral information into a single invariant. For a graph Laplacian $L$ with $l$ distinct eigenvalues, each eigenvalue $\lambda_i$ is associated with a projection matrix onto its eigenspace $\Pi_i = \sum_{j=1}^{J_i} z_{i,j} z_{i,j}^T$, where $\{z_{i,j}\}_{j=1}^{J_i}$ is any orthonormal basis of the eigenspace. Importantly, these projections are basis-independent. The **_eigenspace projection invariant_** is the RPE $\mathcal{P} : V \times V \to \mathbb{R}^{2l}$ defined

$$\mathcal{P}(u,v) = \{(\lambda_i, \Pi_i(u,v)) \, : \, 1 \leq i \leq l\}$$

The WL test using $\mathcal{P}$ as an encoding is called the **_EP-WL test_**. EP-WL is known to be weaker than 3-WL but stronger than the 1-WL test. It is also more powerful than any other spectral test, e.g., the GD-WL tests using spectral distance (Zhang et al., 2024).

Importantly, while EP-WL has been shown to be powerful as a positional encoding, it scales with the size of the graph and as such can be prohibitively large in practice for both preprocessing and training time, incurring $O(n^3)$ space complexity for the full projections. As such, one popular option (Lim et al., 2023; Ma et al., 2023) for employing eigenspace projections in practice is to simply choose the $k$ smallest eigenvalues $\lambda_i$ for the encoding $\mathcal{P}^k(u,v) = \{(\lambda_i, P_i(u,v)) \, : \, 1 \leq i \leq k\}$; we dub this variant the $k$-EP-WL invariant. Importantly, $k$-EP-WL only computes $k$ projections with space complexity $O(kn^2)$. We prove the following about this type of truncation. See Figure 1

**Theorem 4.1.** *There exist a pair of graphs $G$ and $H$, both with $n$ vertices, such that $G$ and $H$ are distinguishable by the 1-WL test, but are indistinguishable by the $k$-EP-WL test for $k \in \Omega(n)$.*

Importantly, this proves that truncated EP-WL test is incomparable to the WL hierarchy, as there are pairs of graphs where it is both stronger and weaker than 1-WL. The proof of Theorem 4.1 is in Appendix A.2. This theorem has important implications for the design of expressive graph transformers. Truncated eigenvectors are one of the most common PEs for graph transformers, but Theorem 4.2 proves that using these encodings alone is not enough to guarantee the transformer is more expressive than MPNNs. This suggests there is real utility in using some form of adjacency information like adjacency PE (Black et al., 2024b), shortest-path distance (Ying et al., 2021; Zhang et al., 2023), or message-passing layers (Rampasek et al., 2022) in addition to eigenvector encodings.

## 4.2. Walk Encodings

Powers of the adjacency matrix count the number of walks between two nodes, where $(A^k)_{uv}$ counts the number of length-$k$ walks between nodes $u$ and $v$. In graph transformers, these matrices are used as positional encodings to inject structural bias into the attention block.[3] We can define $k$-Walk-WL, where the RPE is the first $k$ powers of the adjacency matrix. This generalizes using just the adjacency matrix as an RPE, which has equivalent power as the 1-WL test. It is known that $k$-Walk-WL with $k = \Theta(n)$ is as powerful as WP-EL, meaning it is stronger than 1-WL but weaker than 3-WL; see (Black et al., 2024b, Theorem 4.6) or (Gai et al., 2025, Lemma 3.17).

We show that there are graph pairs that cannot be distinguished by $k$-Walk-WL with a linear number of adjacency powers, but can be distinguished by $k$-EP-WL with $k = 1$

**Theorem 4.2.** *Let $n \geq 6$ even. There exist a pair of graphs $G$ and $H$ with $n$ vertices that are indistinguishable by $\Omega(n)$-Walk-WL but are distinguishable by 1-EP-WL.*

Theorems 4.1 and 4.2 show that the space of truncated PES is more nuanced than the complete PEs (Zhang et al., 2024): no truncated PE is strictly more expressive on its own, and all that we have discussed are capable of being both stronger and weaker than the classic 1-WL or $A$-WL test depending on the forms of truncation used. This motivates our experimental section (Section 5) where we use a fixed combination of different truncated positional encodings, to reinforce the idea that no single truncated encoding on its own is likely to yield the best results.

## 4.3. Effective Resistance

The effective resistance (aka resistance distance) is another well-studied RPE for graph transformers (Zhang et al., 2023) and MPNNs (Velingker et al., 2023). Intuitively, effective resistance measures how well-connected $u$ and $v$ are, or when viewed as a circuit, how much resistance would exist between the nodes. It is also proportional to the commute time between $u$ and $v$ (Chandra et al., 1996).

The **_effective resistance_** between two nodes $u$ and $v$ is.

$$R(s,t) = (1_s - 1_t)^T L^+ (1_s - 1_t)$$

where $1_x$ is an indicator vector that is 1 at index $x$ and zero everywhere else, and $L^+$ is the pseudoinverse of the Laplacian. We call the GD-WL using effective resistance as the RPE the **_Resistance-WL test_**.

We prove several results comparing Resistance-WL to two other WL tests: the adjacency-WL test and the EP-WL test.

---

[3] It is also common to take powers of other graph matrices, like the random walk matrix (Ma et al., 2023; Stoll et al., 2026).

In both cases, we show there are graphs that are indistinguishable by the Resistance-WL test but that can be distinguished by the other tests.

**Theorem 4.3.** *There exist a pair of weighted graphs $G$ and $H$ that are indistinguishable by the Resistance-WL test, but are distinguishable by the Adjacency-WL test.*

The proof of these theorems can be found in Appendix B.

This result is notable for two reasons. First, while we know that the EP-WL test is strictly stronger than the adjacency-WL test, we do not know whether partial spectral invariants like the effective resistance are also stronger than the adjacency-WL test. Moreover, while these results are for weighted graphs, Black et al. (2024b, Proposition 4.10) prove that on unweighted graphs, the adjacency-WL test is equivalent to the the classical 1-WL test. This leads us to conjecture the Resistance-WL test is not stronger than the 1-WL test on unweighted graphs. If this is true, this would disprove a conjecture by Zhang et al. (2023, Section 6) that the Resistance-WL test is stronger than the shortest-path-WL test. Second, as the matrix $A$ can be recovered by the EP-WL test, then we have the following corollary:

**Corollary 4.1.** *There exists a pair of weighted graphs $G$ and $H$ that are indistinguishable by the Resistance-WL test but are distinguishable by the EP-WL test.*

Corollary 4.1 disproves a conjecture by Zhang et al. (2023, Section 6) that the Resistance-WL test encodes the entire spectrum of the graph.

In Appendix B, we discuss the possibility and implications of extending these results to unweighted graphs.

### 4.4. $k$-Harmonic Distances

The results in Section 4.3 proved that Resistance-WL does not encode the entire spectrum of the graph. Given this limitation, we explore a generalization of the effective resistance known as the $k$-harmonic distances (Black et al., 2024a). While the eigenspace projections and walk encodings have natural extensions that encode the entire spectrum of the graph, it is natural to explore this same question for the effective resistance (the 1-harmonic distance).

Previous work has given us reason to believe the $k$-harmonic distances could be useful positional encodings. This work has shown that the biharmonic (2-harmonic) distance encodes structural information about that is different but complementary to the information the effective resistance encodes. While the effective resistance encodes how *well-connected* nodes $s$ and $t$ are, the biharmonic distance encodes how *central* the edge $(s,t)$ is to the graph (Black et al., 2024a), making it a natural candidate to be a PE. Motivated by this intuitive explanation of the biharmonic distance for edges in the graph, we explore the $k$-harmonic dis-

tances and their power in the context of GTs and MPNNs.

The ***k*-harmonic distance** between nodes $u$ and $v$ is[4]

$$H^k(u,v) = \sqrt{(1_u - 1_v)^T (L^+)^k (1_u - 1_v)}$$

As we explore the $k$-harmonic distance as a PE both for GTs and MPNNS, we define a ***Sparse-$\psi$-WL test***, analogous to the $\psi$-WL test, in order to discuss the expressive power of these $k$-harmonic distances for MPNNs. Succinctly, with $\psi$ taken to be the $k$-harmonic distance(s), the sparse-$\psi$-WL test provides an upper bound on the expressive power of MPNNs that use the $k$-harmonic distance as edge features, denoted sparse-$k$-harmonic-WL. We provide a formal definition in Appendix D.1.

**Bounds on $k$-Harmonic-WL**    First, we establish the classic bounds that are common of any spectral distance. We show that any Sparse-$k$-Harmonic-WL test yields a PE that is stronger that 1-WL; in short, any Sparse-WL test will be stronger than the 1-WL test as it incorporates edge information. The same cannot be said of the $k$-Harmonic-WL test. While we do not know whether or not this is the case for unweighted graphs, Theorem 4.3 shows the analogous theorem for weighted graph is not true. Additionally, we show that both the $k$-harmonic distance and Sparse-$k$-harmonic WL tests are weaker than 3-WL.

**Theorem 4.4.** *Let $k \geq 1$. The Sparse-$k$-Harmonic-WL test is strictly stronger than the WL test.*

**Theorem 4.5.** *The $3$-WL test is strictly stronger than the $k$-Harmonic-WL test and Sparse-$k$-Harmonic-WL tests for all $k \in \mathbb{R}$.*

Proofs are found in Appendices D.2 and D.3.

The EP-WL and $\Theta(n)$-Walk WL tests have equivalent expressive power, and both lie between the 1-WL and 3-WL tests. To complement these results, we prove that concatenating multiple $k$-harmonic distances gives a RPE that is equivalent in power to these two encodings. For a set of values $S \subset \mathbb{R}$, let the $S$-harmonic WL test denote the $\psi$-WL test for $\psi : V \times V \to \mathbb{R}^{|S|}$ defined $\psi(u,v) = (H^{(k)}(u,v) : k \in S)$, i.e., we concatenate the $k$-harmonic distances for all $k \in S$. We can prove that $[2n]$-harmonic distances are equal in power to the EP-WL test.

**Theorem 4.6.** *Let $[2n] = \{1, \ldots, 2n\}$. The $[2n]$-harmonic WL test is as strong as the EP-WL test*

This mirrors known results for other spectral PEs like heat kernels (Black et al., 2024b, Theorem 4.9).

---

[4]The effective resistance is defined without the square root. Surprisingly, even without the square root, the effective resistance is a metric (Klein & Randić, 1993).

**Comparing Different $k$-Harmonic Distances**   Previous sections show that the truncated walk and projection-based PEs provably capture specific aspects of graph structure. While there is also a "complete" PE using the $k$-harmonic distances (Theorem 4.6), we also want to understand the strength of truncated $k$-harmonics. In particular, prior research (Black et al., 2024a) show that the 1- and 2-harmonic have very different properties and interpretations, so we ask whether this continues to be true when the $k$-harmonic distances are used as PEs. On one side, we will show that the $k$-harmonic distance capture different information about a graph, as there are graphs that the 2-harmonic can distinguish in significantly fewer iterations that the effective resistance (Theorem 4.7). However, on the other side, we show that if one $k$-harmonic distance can distinguish a pair of graphs, then most $k$-harmonic distances will distinguish these graphs as well.

**Theorem 4.7.** *There are pairs of graphs that Sparse-Biharmonic-WL can distinguish in one iteration but Sparse-Resistance-WL cannot distinguish in $o(n)$ iterations.*

The pairs of graphs we consider in this proof are both trees. To prove this theorem, we will use the following fact.

**Lemma 4.1.** *The Sparse-Resistance-WL test is equally strong as the WL test when $G$ and $H$ are trees*

This result is directly implied by a result of Ghosh et al. (2008, Theorem 2.3) that the effective resistance between any two nodes in a tree is their shortest path distance. Thus, the effective resistance of all edges in a tree is 1, giving no additional information. Conversely, Black et al. (2024a, Theorem 5.1) proves that the biharmonic distance for any edge in a tree entirely determined by the number of nodes on either side of that edge, which supplies additional topological information about an edge. The full proof of Theorem 4.7 is contained in Appendix D.4. While this may suggest that Sparse-Biharmonic-WL is much more powerful than Sparse-Resistance-WL on trees, this does not generalize to all trees. We can construct a counterexample consisting of a pair of non-isomorphic trees that Sparse-Biharmonic-WL cannot distinguish in $o(n)$ iterations. We defer this example to Appendix D.5.

These findings formally demonstrate a gap in expressive power among $k$-harmonics: the biharmonic distance reflects global graph structure, but effective resistance can be insufficient in particular cases.

While the previous result proves different $k$-harmonic distances can have different powers, the following theorem proves that this is generally not the case: if one $k$-harmonic distance can be used to distinguish a pair of graphs, then most other $k$-harmonic distances will distinguish these graphs as well.

**Theorem 4.8.** *Let $G$ and $H$ be graphs with $n$ vertices that are distinguishable by $k$-Harmonic-WL or Sparse-$k$-Harmonic-WL for some $k$. Then for all but $O(n^5)$ values of $k' \in \mathbb{R}^+$, $G$ and $H$ are distinguishable by the $k'$-harmonic-WL test or test Sparse-$k'$-harmonic WL test respectively.*

**Conclusion**   In summary, this section makes two contributions. First, we establish several positive results showing that the $k$-harmonic distances form a viable family of PEs, naturally interpreted as truncations with intuitive meanings. We also show the existence of a complete $k$-harmonic representation that complements these truncations. Second, we prove an upper bound on the power of truncated $k$-harmonics, paralleling limits observed for other popular PEs. These findings motivate our experiments: we (i) evaluate how $k$-harmonics behave empirically and (ii) show that mixtures of PEs generally perform best, reflecting the negative results from Section 4.

**Runtime**   While the naive algorithm for computing $k$-harmonic distances take $O(n^3)$ time, there is an algorithm to approximately compute any $k$-harmonic distance that is linear in $k$ and the number of edges $m$ in the graph. This is a particularly relevant result given the motivation behind using truncated positional encodings to begin with—to provide structural information without incurring large computational overhead and training costs. The algorithm avoids exact computation of the pseudoinverse of the Laplacian using the fast Laplacian solvers and Johnson-Lindenstrauss projections (Johnson & Lindenstrauss, 1984). We can approximately compute all pairs $k$-harmonic distance in $O(mk \operatorname{poly} \log n + n^2 \log n)$ time and the $k$-harmonic distance on the edges in $O(mk \operatorname{poly} \log n)$ time. We outline this algorithm in Appendix D.8.

## 5. Experiments

As an exploration of our theoretical results, we test several different versions of truncated PEs. That is, given the limited expressivity of any singular truncated PE, we experiment with concatenating them together to see whether or not it yields increased performance. Further, we conduct experiments specifically for the $k$-harmonics in order to explore their usefulness as a PE. We provide further experimentation in Appendix E.3

**Architecture**   We use the Graphormer-GD (Zhang et al., 2023) architecture as outlined in Section 3. For eigenspace projections we use a simple 2-layer MLP or a DeepSets (Zaheer et al., 2018) module as choice of function $\phi$ before they are injected into the attention layer. For some experiments we opt to use MPNNs, for which PEs are given as edge features with the GINE architecture (Hu et al., 2020). We provide further experimental settings in Appendix E.

*Table 1.* % Accuracy for each family of graphs in BREC.

| Accuracy | $k$-harmonics | | | $k$-EP-WL | | | | | | 3-WL |
|---|---|---|---|---|---|---|---|---|---|---|
| | Resistance | Biharmonic | 4-harmonic | $k=1$ | $k=2$ | $k=3$ | $k=4$ | $k=5$ | $k=7$ | - |
| **Basic** | 100 | 100 | 100 | 0 | 46.6 | 86.7 | 90 | 90 | 91.6 | 100 |
| **Regular** | **35.7** | 35 | 30.7 | 0 | 27.1 | 32.1 | 31.4 | 32.1 | 30.7 | 35.7 |
| **Extension** | 100 | **100** | 97 | 0 | 59 | 83 | 88 | 91 | 92 | 100 |
| **CFI** | 7 | **11** | 4 | 3 | 3 | 2 | 1 | 2 | 2 | 60 |
| **Total** | 54.25 | **55** | 51 | 0 | 32 | 46.5 | 46.75 | 47 | 48 | 0 |

## 5.1. BREC

The BREC dataset was introduced by (Wang & Zhang, 2024) as an means of measuring the *realized* expressivity of GNNs. The dataset uses a contrastive learning approach to test whether a GNN is able to learn to map non-isomorphic graphs to different features in latent space. The dataset consists of several different types of graphs that range from WL indistinguishable to 4-WL indistinguishable.

We test our theoretical results on the truncation of PEs by testing "how much information" is needed before a PE is able to return to its complete baseline. In all tests, we would expect the complete forms of the PEs to distinguish all WL indistinguishable graphs and none of the 3-WL indistinguishable graphs (as per their expressivity bounds). Further, we test with the transformer architecture and include dense PEs (all pairs of nodes have an associated value).

We summarize the main results in Table 1. In $k$-EP-WL, increasing $k$ linearly increases the dimensionality of the PE per edge, whereas the $k$-harmonics remain one-dimensional regardless of the chosen $k$. We therefore do not concatenate multiple $k$-harmonics the way we concatenate eigenspace projections: on BREC, a single $k$-harmonic already attains its theoretical ceiling on the Basic, Regular, and Extension graphs — separating all pairs of graphs between 1-WL and 3-WL—while it performs worse than the 3-WL test on the CFI graphs. Adding more eigenspace projections yields a sharp jump from $k=2$ to $k=3$, with diminishing gains thereafter. Taken together, these findings argue for the practicality of $k$-harmonic distances: they provide competitive results through a single-channel with lower computational overhead and training time. The results for 3-WL are from the original paper (Wang & Zhang, 2024). We provide further explanation and experimentation on the $k$-harmonics in Appendix E.1

Lastly, we note that the powers of the adjacency matrix are already well-studied on BREC with similar experimental settings (Black et al., 2024b). For summary, stacking 20 powers of the adjacency matrix yields a total score of 53.5%, competitive with the biharmonic distance, but similarly is a multi-dimensional PE.

## 5.2. ZINC

While our previous analysis focuses on theoretical expressivity for graph isomorphism, our motivation is practical: to chart the space of PEs that practitioners actually use. To that end, we evaluate on ZINC-12k—a graph regression benchmark introduced by (Dwivedi et al., 2023) for predicting molecular constrained solubility — to test the conjecture that combining multiple truncated PEs outperforms any single PE.

We fix the dimensionality of PEs to 8 (not counting the edge features supplied by ZINC) and test them as equal mixtures. Projection refers to keeping the $k$ smallest eigenvalues and using their eigenspace projections. Walk refers to $k$ powers of the adjacency matrix concatenated together. $k$-harmonics refers to that number of harmonics concatenated together. We present these results in Table 2 and find that the mixture of truncated walks and $k$-harmonics achieves the best performance, surpassing either truncated PE in isolation.

Moreover, mixing PEs helps across the board except for the pairing of projections with $k$-harmonics. That combination outperforms projections alone, but $k$-harmonics achieve essentially the same performance with or without projections. This suggests that the inductive biases provided by truncated projections may be redundant in presence of the information provided by the $k$-harmonics.

## 6. Conclusion

In this paper, we compared three families of relative positional encodings for graph neural networks: eigenspace projections, walk matrices, and $k$-harmonic distances. While these three families have equivalent expressive power in the complete forms, we prove that they have very different expressive power in their truncated forms. Although the applicability of the graph isomorphism decision problem has shown to be limited on real world data, we ar-

*Table 2.* Test MAE for different PE combinations on ZINC-12k. Lower is better.

| | Projections | $k$-Harmonics | Walks | Test MAE |
|---|---|---|---|---|
| | Dimensionality of Encoding | | | |
| **Projections** | 8 | – | – | $0.117 \pm 0.005$ |
| $k$**-harmonics** | – | 8 | – | $0.076 \pm 0.006$ |
| **Walks** | – | – | 8 | $0.082 \pm 0.003$ |
| **Projections + Walks** | 4 | – | 4 | $0.075 \pm 0.005$ |
| **Projections +** $k$**-harmonics** | 4 | 4 | – | $0.078 \pm 0.002$ |
| **Walks +** $k$**-harmonics** | – | 4 | 4 | $\mathbf{0.064 \pm 0.002}$ |

gue that it remains a principled proxy for the kinds of structural signals a positional encoding can expose to a model. In this sense, our results help explain why truncated PEs may behave differently in downstream tasks, even when their full counterparts may be theoretically more expressive.

Based on this observation, we recommend that practitioners consider combining encodings from multiple truncated families in order to capture complementary structural cues while controlling preprocessing and training cost. At the same time, our conclusions should be interpreted with appropriate scope: the primary contribution of this work is to initiate the theoretical study of truncated positional encodings and their combinations. We hope this work provides both a foundation for understanding truncated PEs and a starting point for more systemic empirical study in the future. More broadly, our results suggest that truncated positional encodings should be viewed as a distinct design space in their own right, rather than merely as approximations of their complete counterparts.

## Impact Statement

This paper presents work whose goal is to advance the field of Machine Learning. There are many potential societal consequences of our work, none which we feel must be specifically highlighted here.

## Funding

Mitchell Black is supported by NSF grant CCF-2217058. Amir Nayyeri is supported by NSF grants CCF-1941086 and CCF-2311180.

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

# A. Truncated EP-WL is Not Stronger than the WL Test.

In this section, we will prove that there are graphs that are distinguishable by the WL test, but are indistinguishable by the truncated EP-WL test when we use a constant fraction of the eigenpairs.

## A.1. Preliminaries

Let $G$ and $H$ be graphs. The ***product graph*** of $G$ and $H$ is the graph $G \times H$ with vertices $V_{G \times H} = V_G \times V_H$ and edge $E_{G \times H} = \{((u_1, v_1), (u_2, v_2)) : (u_1, u_2) \in E_G \text{ or } (v_1, v_2) \in E_H\}$. The eigenvectors of product graphs have the following property:

**Lemma A.1** ((Spielman, 2025, Theorem 5.3.2)). *Let $G$ and $H$ be graphs, and let $\{(\lambda_{G,i}, x_{G,i} : 1 \leq i \leq V_G\}$ and $\{(\lambda_{H,j}, x_{H,j}) : 1 \leq j \leq |V_H|\}$ be the eigenpairs of $G$ and $H$ respectively. Then the product graph $G \times H$ has eigenpairs $\{(\lambda_{G \times H,(i,j)}, x_{G \times H,(i,j)} : 1 \leq i \leq |V_G|, 1 \leq j \leq |V_H|\}$ where*

$$\lambda_{G \times H,(i,j)} = \lambda_{G,i} + \lambda_{H,j}$$

*and*

$$x_{G \times H,(i,j)}(u, v) = x_G(u) \cdot x_H(v) \qquad\qquad (u \in V_G \text{ and } v \in V_H)$$

*Alternatively, $x_{G \times H,(i,j)} = x_{G,i} \otimes x_{H,j}$, where $\otimes$ is the Kronecker product.*

The following lemmas describe the spectra of several families of graphs we will use in our proof.

**Lemma A.2** ((Spielman, 2025, Theorem 5.6.1)). *Let $P_n$ be the path graph on $n$ vertices. The eigenvalues of the Laplacian of $P_n$ are $2 - 2\cos(\pi i/n)$ for $0 \leq i < n$.*

**Lemma A.3** ((Spielman, 2025, Lemma 5.1.1)). *Let $K_n$ be the complete graph on $n$ vertices. The eigenvalues of the Laplacian of $K_n$ are $0$ with multiplicity $1$ and $n$ with multiplicity $n - 1$*

**Lemma A.4** ((Spielman, 2025, Section 29.1)). *Let $K_{n,n}$ be the complete bipartite graph between two sets of $n$ vertices. The eigenvalues of the Laplacian of $K_{n,n}$ are $0$ with multiplicity $1$, $n$ with multiplicity $2n - 2$, and $2n$ with multiplicity $1$.*

## A.2. Proof of Theorem 4.1

**Theorem 4.1.** *There exist a pair of graphs $G$ and $H$, both with $n$ vertices, such that $G$ and $H$ are distinguishable by the 1-WL test, but are indistinguishable by the $k$-EP-WL test for $k \in \Omega(n)$.*

*Proof.* We will prove this for the graphs $G = P_n \times K_{10}$ and $H = P_n \times K_{5,5}$ for any $n \geq 2$.

First, observe that $P_n \times K_{10}$ and $P_n \times K_{5,5}$ are distinguishable by the WL-test as all nodes in $P_n \times K_{10}$ have degree $\geq 9$, while all nodes of $P_n \times K_{5,5}$ have degree $\leq 7$.

Next, we will prove that $P_n \times K_{10}$ and $P_n \times K_{5,5}$ are indistinguishable by $k$-EP-WL for any value of $k \leq n$.

First, by Lemma A.2, observe that all eigenvalues $\lambda_i$ of $P_n$ satisfy $\lambda_i = 2 - 2\cos(\pi i/n) < 5$, while all non-zero eigenvalues of both $K_{10}$ and $K_{5,5}$ are $\geq 5$ by Lemmas A.3 and A.4; thus, all eigenvalues of $P_n$ are smaller than all non-zero eigenvalues of both $K_{10}$ and $K_{5,5}$. This implies the product graphs $P_n \times K_{10}$ and $P_n \times K_{5,5}$ have the same smallest $n$ eigenvalues $\lambda_{P_n \times K_{10},(i,0)} = \lambda_{P_n,i} = \lambda_{P_n \times K_5,(i,0)} < 5$, i.e., the eigenvalues of $P_n$ plus the zero eigenvalue of $K_{10}$ or $K_{5,5}$, as any eigenvalue $\lambda_{P_n, K_n,(i,j)}$ or $\lambda_{P_n, K_n,(i,j)}$ for $j \geq 1$ will be $\geq 5$.

Likewise, $P_n \times K_{10}$ and $P_n \times K_{5,5}$ have the same smallest $n$ eigenvectors (for a specific choice of basis), as $K_{10}$ and $K_{5,5}$ have the same 0th eigenvector (the length-10 all-ones vectors $1_{10}$). By "same", we mean that there are choices of eigenvectors of $P_n \times K_{10}$ and $P_n \times K_{5,5}$ and a bijection $\sigma : V_{P_n \times K_{10}} \to V_{P_n \times K_{5,5}}$ such that $x_{P_n \times K_{10},i}(v) = x_{P_n \times K_{5,5}}(\sigma(v))$. Pick an arbitrary bijection $\tau : K_{10} \to K_{5,5}$, and define $\sigma : V_{P_n \times K_{10}} \to V_{P_n \times K_{5,5}}$ as $\sigma(u, v) = (u, \sigma(v))$, i.e., each vertex in the one copy of $P_n$ in $P_n \times K_{10}$ to a vertex in the same copy of $P_n$ in $P_n \times K_{5,5}$. For any choice of orthonormal eigenvectors $x_{P_n,i}$ of $P_n$, then by Lemma A.1, the first $n$ eigenvectors of $P_n \times K_{10}$ and $P_n \times K_{5,5}$ are $x_{P_n,i} \otimes 1_{10}$. It is easy to see that $P_n \times K_{10}$ and $P_n \times K_{5,5}$ have the same entries in $x_{P_n,i} \otimes 1_{10}$ for the bijection defined above. See Figure 1. Moreover, for the projection onto this eigenspace $\Pi_i$, we can see that $\Pi_i(u, v) = x_{P_n \times K_{10},i}(u) \cdot x_{P_n \times K_{5,5},i}(u) = \Pi_i(\sigma(u), \sigma(v))$ for $u, v \in V_{P_n \times K_{10}}$.

Informally, because $P_n \times K_{10}$ and $P_n \times K_{5,5}$ have the same first $n$ eigenvalues and eigenvectors, they are indistinguishable by $n$-EP-WL.

To make this more formal, we must show that $P_n \times K_{10}$ and $P_n \times K_{5,5}$ have the same set of $k$-EP-WL colors at each iteration. To do this, we need a mapping between the vertices of $P_n \times K_{10}$ and $P_n \times K_{5,5}$ that preserves colors. We can do this with the bijection $\sigma$ defined above, and we can inductively prove that $\chi^{(t)}_{P_n \times K_{10}}(v) = \chi^{(t)}_{P_n \times K_{5,5}}(v)$ for all $v \in V_{P_n \times K_{10}}$ and $t \geq 0$ as this bijection preserves entries in the eigenspace projection matrices $\Pi_i$. $\qquad\square$

**Corollary A.1.** *There exist a pair of graphs $G$ and $H$, both with $n$ vertices, such that $G$ and $H$ are distinguishable by the $l$-RWPE test for all $l \geq 1$, but are indistinguishable by the $k$-EP-WL test for $k \in \Omega(n)$.*

# B. Weighted Resistance-WL is Not Stronger than Weighted WL

In this section, we prove several results comparing the resistance WL test to two other WL tests: the adjacency-WL test and the EP-WL-test. In both cases, we show there are graphs that are indistinguishable by the resistance WL test but that can be distinguished by the other other tests.

### B.1. Proof of Theorem 4.3

**Theorem 4.3.** *There exist a pair of weighted graphs $G$ and $H$ that are indistinguishable by the Resistance-WL test, but are distinguishable by the Adjacency-WL test.*

*Proof.* The outline of our proof is as follows.

1. We construct a pair of squared Euclidean distances matrices and prove they are indistinguishable by the WL test.

2. We prove that these distance matrices are the effective resistance matrices of a pair of weighted graphs.

3. We prove that these two graphs are indistinguishable by the WL test on their weighted adjacency matrices.

We are able to prove Item 1 analytically. For Items 2 and 3, we use symbolic software to perform a computer-aided proof. These steps involving taking a matrix pseudoinverse, which can be hard to reason about mathematically but can by checked with a simple Python program using the SymPy (Meurer et al., 2017) library to perform exact rational operation. The code for this proof can be found in the accompanying GitHub repo.

To prove Item 1, we define point sets $P = \{p_1, \ldots, p_{10}\}$ and $Q = \{q_1, \ldots, q_{10}\}$.[5] When represented as matrices with columns $p_i$, these point sets are

$$P = \frac{1}{10} \begin{bmatrix} 8 & 1 & 0 & 0 & 0 & 0 & 0 & 0 & 0 & 1 \\ 1 & 8 & 1 & 0 & 0 & 0 & 0 & 0 & 0 & 0 \\ 0 & 1 & 8 & 1 & 0 & 0 & 0 & 0 & 0 & 0 \\ 0 & 0 & 1 & 8 & 1 & 0 & 0 & 0 & 0 & 0 \\ 0 & 0 & 0 & 1 & 8 & 1 & 0 & 0 & 0 & 0 \\ 0 & 0 & 0 & 0 & 1 & 8 & 1 & 0 & 0 & 0 \\ 0 & 0 & 0 & 0 & 0 & 1 & 8 & 1 & 0 & 0 \\ 0 & 0 & 0 & 0 & 0 & 0 & 1 & 8 & 1 & 0 \\ 0 & 0 & 0 & 0 & 0 & 0 & 0 & 1 & 8 & 1 \\ 1 & 0 & 0 & 0 & 0 & 0 & 0 & 0 & 1 & 8 \end{bmatrix}$$

---

[5]Looking at the matrices of these point sets, readers familiar with the WL test might be reminded of the classic example of two graphs that are indistinguishable by the WL test: the 10-cycle $C_{10}$ and the disjoint union of two five cycles $C_5 \sqcup C_5$. Indeed, this is how we constructed these points. We found these point sets using a construction by Maehara (1984) to find point sets whose unit-disk graphs is a given graph, and the unit-disk graphs of these point sets are $C_{10}$ and $C_5 \sqcup C_5$.

$$Q = \frac{1}{10} \begin{bmatrix} 8 & 1 & 0 & 0 & 1 & 0 & 0 & 0 & 0 & 0 \\ 1 & 8 & 1 & 0 & 0 & 0 & 0 & 0 & 0 & 0 \\ 0 & 1 & 8 & 1 & 0 & 0 & 0 & 0 & 0 & 0 \\ 0 & 0 & 1 & 8 & 1 & 0 & 0 & 0 & 0 & 0 \\ 1 & 0 & 0 & 1 & 8 & 0 & 0 & 0 & 0 & 0 \\ 0 & 0 & 0 & 0 & 0 & 8 & 1 & 0 & 0 & 1 \\ 0 & 0 & 0 & 0 & 0 & 1 & 8 & 1 & 0 & 0 \\ 0 & 0 & 0 & 0 & 0 & 0 & 1 & 8 & 1 & 0 \\ 0 & 0 & 0 & 0 & 0 & 0 & 0 & 1 & 8 & 1 \\ 0 & 0 & 0 & 0 & 0 & 1 & 0 & 0 & 1 & 8 \end{bmatrix}$$

If we compute squared Euclidean distance matrices of these point sets, we find that all rows of both matrices have the same multiset of distances; one 0, two 1s, two 1.30s, and five 1.32s. Therefore, if we perform GD-WL on the two matrices, then at each iteration, all nodes in both graphs will have the same color. Therefore, these two points are indistinguishable by the RPE-WL test.

$$d(P,P)^2 = \begin{bmatrix} 0.00 & 1.00 & 1.30 & 1.32 & 1.32 & 1.32 & 1.32 & 1.32 & 1.30 & 1.00 \\ 1.00 & 0.00 & 1.00 & 1.30 & 1.32 & 1.32 & 1.32 & 1.32 & 1.32 & 1.30 \\ 1.30 & 1.00 & 0.00 & 1.00 & 1.30 & 1.32 & 1.32 & 1.32 & 1.32 & 1.32 \\ 1.32 & 1.30 & 1.00 & 0.00 & 1.00 & 1.30 & 1.32 & 1.32 & 1.32 & 1.32 \\ 1.32 & 1.32 & 1.30 & 1.00 & 0.00 & 1.00 & 1.30 & 1.32 & 1.32 & 1.32 \\ 1.32 & 1.32 & 1.32 & 1.30 & 1.00 & 0.00 & 1.00 & 1.30 & 1.32 & 1.32 \\ 1.32 & 1.32 & 1.32 & 1.32 & 1.30 & 1.00 & 0.00 & 1.00 & 1.30 & 1.32 \\ 1.32 & 1.32 & 1.32 & 1.32 & 1.32 & 1.30 & 1.00 & 0.00 & 1.00 & 1.30 \\ 1.30 & 1.32 & 1.32 & 1.32 & 1.32 & 1.32 & 1.30 & 1.00 & 0.00 & 1.00 \\ 1.00 & 1.30 & 1.32 & 1.32 & 1.32 & 1.32 & 1.32 & 1.30 & 1.00 & 0.00 \end{bmatrix}$$

$$d(Q,Q)^2 = \begin{bmatrix} 0.00 & 1.00 & 1.30 & 1.30 & 1.00 & 1.32 & 1.32 & 1.32 & 1.32 & 1.32 \\ 1.00 & 0.00 & 1.00 & 1.30 & 1.30 & 1.32 & 1.32 & 1.32 & 1.32 & 1.32 \\ 1.30 & 1.00 & 0.00 & 1.00 & 1.30 & 1.32 & 1.32 & 1.32 & 1.32 & 1.32 \\ 1.30 & 1.30 & 1.00 & 0.00 & 1.00 & 1.32 & 1.32 & 1.32 & 1.32 & 1.32 \\ 1.00 & 1.30 & 1.30 & 1.00 & 0.00 & 1.32 & 1.32 & 1.32 & 1.32 & 1.32 \\ 1.32 & 1.32 & 1.32 & 1.32 & 1.32 & 0.00 & 1.00 & 1.30 & 1.30 & 1.00 \\ 1.32 & 1.32 & 1.32 & 1.32 & 1.32 & 1.00 & 0.00 & 1.00 & 1.30 & 1.30 \\ 1.32 & 1.32 & 1.32 & 1.32 & 1.32 & 1.30 & 1.00 & 0.00 & 1.00 & 1.30 \\ 1.32 & 1.32 & 1.32 & 1.32 & 1.32 & 1.30 & 1.30 & 1.00 & 0.00 & 1.00 \\ 1.32 & 1.32 & 1.32 & 1.32 & 1.32 & 1.00 & 1.30 & 1.30 & 1.00 & 0.00 \end{bmatrix}$$

We now prove Item 2, where we verify this set of squared Euclidean distances are the effective resistances of a weighted graph. For this, we rely on the work of Devriendt (2022). Devriendt (2022, Proposition 1) proves that a matrix is a Laplacian of a weighted graph iff (1) it is symmetric, (2) it has non-positive off-diagonal entries, and (3) its rows and columns sum to 0.[6] Likewise, Devriendt (2022, Proposition 2) proves a squared Euclidean distance matrix is the resistance matrix of a weighted graph iff the pseudoinverse of its Gram matrix is a weighted Laplacian. Finally, given a squared Euclidean distance matrix $D$, we can recover its Gram matrix with double-centering: the Gram matrix is $-\frac{1}{2}\Pi D\Pi$, where $\Pi = I - \frac{1}{n}J$ for $J$ the all-ones matrix. Therefore, to verify that these squared Euclidean distance matrices are the effective resistance matrix of a graph, we convert them to their Gram matrices, take the pseudoinverse of the Gram matrices, and verify they satisfy the three conditions to be a Laplacian. We do this in Python using the SymPy library. Sympy provides exact calculation using rational numbers, so these calculations (including taking the pseudoinverse) are not subject to numerical errors.

Finally, we only need to verify that these graphs can be distinguished by RPE-WL on their weighted adjacency matrices. As mentioned in the previous paragraph, we can exactly compute the adjacency matrices of these graphs using SymPy. The

---

[6]Devriendt's fourth condition that the matrix be irreducible is only for *connected* graphs. For our purposes, we do not care whether or not our graphs are connected, although it turns out that they are connected.

adjacency matrices are:

$$A_P = \begin{bmatrix}
0 & \frac{5822669}{10979298} & \frac{202579}{10979298} & \frac{1249229}{10979298} & \frac{1073419}{10979298} & \frac{1105229}{10979298} & \frac{1073419}{10979298} & \frac{1249229}{10979298} & \frac{202579}{10979298} & \frac{5822669}{10979298} \\
\frac{5822669}{10979298} & 0 & \frac{5822669}{10979298} & \frac{202579}{10979298} & \frac{1249229}{10979298} & \frac{1073419}{10979298} & \frac{1105229}{10979298} & \frac{1073419}{10979298} & \frac{1249229}{10979298} & \frac{202579}{10979298} \\
\frac{202579}{10979298} & \frac{5822669}{10979298} & 0 & \frac{5822669}{10979298} & \frac{202579}{10979298} & \frac{1249229}{10979298} & \frac{1073419}{10979298} & \frac{1105229}{10979298} & \frac{1073419}{10979298} & \frac{1249229}{10979298} \\
\frac{1249229}{10979298} & \frac{202579}{10979298} & \frac{5822669}{10979298} & 0 & \frac{5822669}{10979298} & \frac{202579}{10979298} & \frac{1249229}{10979298} & \frac{1073419}{10979298} & \frac{1105229}{10979298} & \frac{1073419}{10979298} \\
\frac{1073419}{10979298} & \frac{1249229}{10979298} & \frac{202579}{10979298} & \frac{5822669}{10979298} & 0 & \frac{5822669}{10979298} & \frac{202579}{10979298} & \frac{1249229}{10979298} & \frac{1073419}{10979298} & \frac{1105229}{10979298} \\
\frac{1105229}{10979298} & \frac{1073419}{10979298} & \frac{1249229}{10979298} & \frac{202579}{10979298} & \frac{5822669}{10979298} & 0 & \frac{5822669}{10979298} & \frac{202579}{10979298} & \frac{1249229}{10979298} & \frac{1073419}{10979298} \\
\frac{1073419}{10979298} & \frac{1105229}{10979298} & \frac{1073419}{10979298} & \frac{1249229}{10979298} & \frac{202579}{10979298} & \frac{5822669}{10979298} & 0 & \frac{5822669}{10979298} & \frac{202579}{10979298} & \frac{1249229}{10979298} \\
\frac{1249229}{10979298} & \frac{1073419}{10979298} & \frac{1105229}{10979298} & \frac{1073419}{10979298} & \frac{1249229}{10979298} & \frac{202579}{10979298} & \frac{5822669}{10979298} & 0 & \frac{5822669}{10979298} & \frac{202579}{10979298} \\
\frac{202579}{10979298} & \frac{1249229}{10979298} & \frac{1073419}{10979298} & \frac{1105229}{10979298} & \frac{1073419}{10979298} & \frac{1249229}{10979298} & \frac{202579}{10979298} & \frac{5822669}{10979298} & 0 & \frac{5822669}{10979298} \\
\frac{5822669}{10979298} & \frac{202579}{10979298} & \frac{1249229}{10979298} & \frac{1073419}{10979298} & \frac{1105229}{10979298} & \frac{1073419}{10979298} & \frac{1249229}{10979298} & \frac{202579}{10979298} & \frac{5822669}{10979298} & 0
\end{bmatrix}$$

$$A_Q = \begin{bmatrix}
0 & \frac{639}{1210} & \frac{39}{1210} & \frac{39}{1210} & \frac{639}{1210} & \frac{1}{10} & \frac{1}{10} & \frac{1}{10} & \frac{1}{10} & \frac{1}{10} \\
\frac{639}{1210} & 0 & \frac{639}{1210} & \frac{39}{1210} & \frac{39}{1210} & \frac{1}{10} & \frac{1}{10} & \frac{1}{10} & \frac{1}{10} & \frac{1}{10} \\
\frac{39}{1210} & \frac{639}{1210} & 0 & \frac{639}{1210} & \frac{39}{1210} & \frac{1}{10} & \frac{1}{10} & \frac{1}{10} & \frac{1}{10} & \frac{1}{10} \\
\frac{39}{1210} & \frac{39}{1210} & \frac{639}{1210} & 0 & \frac{639}{1210} & \frac{1}{10} & \frac{1}{10} & \frac{1}{10} & \frac{1}{10} & \frac{1}{10} \\
\frac{639}{1210} & \frac{39}{1210} & \frac{39}{1210} & \frac{639}{1210} & 0 & \frac{1}{10} & \frac{1}{10} & \frac{1}{10} & \frac{1}{10} & \frac{1}{10} \\
\frac{1}{10} & \frac{1}{10} & \frac{1}{10} & \frac{1}{10} & \frac{1}{10} & 0 & \frac{639}{1210} & \frac{39}{1210} & \frac{39}{1210} & \frac{639}{1210} \\
\frac{1}{10} & \frac{1}{10} & \frac{1}{10} & \frac{1}{10} & \frac{1}{10} & \frac{639}{1210} & 0 & \frac{639}{1210} & \frac{39}{1210} & \frac{39}{1210} \\
\frac{1}{10} & \frac{1}{10} & \frac{1}{10} & \frac{1}{10} & \frac{1}{10} & \frac{39}{1210} & \frac{639}{1210} & 0 & \frac{639}{1210} & \frac{39}{1210} \\
\frac{1}{10} & \frac{1}{10} & \frac{1}{10} & \frac{1}{10} & \frac{1}{10} & \frac{39}{1210} & \frac{39}{1210} & \frac{639}{1210} & 0 & \frac{639}{1210} \\
\frac{1}{10} & \frac{1}{10} & \frac{1}{10} & \frac{1}{10} & \frac{1}{10} & \frac{639}{1210} & \frac{39}{1210} & \frac{39}{1210} & \frac{639}{1210} & 0
\end{bmatrix}$$

We can see that these graph can be distinguished in a single iteration of Adjacency-WL because (for example) the second graph has an edge with weight $\frac{1}{10}$ and the first graph has no edge with this weight. $\square$

**Theorem B.1.** *There exist a pair of weighted graphs $G$ and $H$ such that $G$ and $H$ are indistinguishable by the resistance WL test, but are distinguishable by the EP-WL test.*

*Proof.* As the adjacency matrix can be computed from the EP-WL colorings, then it follows that the EP-WL test is stronger than the resistance WL test. See (Zhang et al., 2024, Proposition 4.2). $\square$

### B.2. Future Directions

**Unweighted Graphs** As mentioned in the motivation of this section, these results are for weighted graphs. It is an interesting open question if similar results hold for unweighted graphs.

The current results would seem to rule out the possibility that the WL iterations could recover the Laplacian from the resistance matrix using only linear algebra. Instead, any proof that resistance WL is stronger than WL on unweighted graphs would have to use some property of resistance that only holds on unweighted graphs, which admittedly, there are quite a few. However, our conjecture is this is not the case.

**Conjecture B.1.** *There exist a pair of unweighted graphs $G$ and $H$ such that $G$ and $H$ are indistinguishable by the resistance WL test, but they are distinguishable by the WL test.*

As the shortest-path-distance WL test is stronger than the WL test, then Conjecture B.1 would imply the following conjecture. However, proving Conjecture B.2 would still be significant as it would disprove a conjecture by (Zhang et al., 2023, Section 6) and would prove that resistance-WL is strong than shortest-path-distance-WL.

**Conjecture B.2.** *There exist a pair of unweighted graphs $G$ and $H$ such that $G$ and $H$ are indistinguishable by the resistance WL test, but they are distinguishable by the shortest-path-distance WL test.*

## C. Truncated Walk Encodings

**Lemma C.1** ((Lehman et al., 2017, Corollary 10.3.3)). *Let $G = (V, E)$ be a graph, let $A$ be the adjacency matrix of $G$, and let $u, v \in V$. Then $A^k_{vu}$ is the number of length $k$ walks starting at $v$ and ending at $u$.*

**Lemma C.2.** *Let $G$ and $H$ be graphs. Let $v \in V_G$ and $u \in V_H$ such that $v$ and $u$ have have isomorphic $k$ hop neighborhoods, and let $\sigma : B_k(v) \to B_k(u)$ be the isomorphism. Then for all $w \in V_{B_k(v)}$ and all $l \le k$, $A_{G,vw}^l = A_{H,u\sigma(w)}^l$.*

*Proof.* Note that any length $l$ starting at $v$ must necessarily be contained in the $k$-hop neighborhood of $v$. Thus, for any $w \in B_k(v)$, $v$ and $w$ and $u$ and $\sigma(w)$ will be connected by the same number of length-$l$ walks as $B_k(v)$ and $B_k(u)$ are isomorphic. The lemma follows by Lemma C.1. □

**Theorem C.1.** *Let $n \ge 6$ be any even integer. There exist a pair of graphs $G$ and $H$ on $n$ vertices that are indistinguishable by $\Omega(n)$-Walk-WL but are distinguishable by 1-EP-WL.*

*Proof.* The graphs $G$ and $H$ are the length-$n$ cycle and the disjoint union of 2 length-$\frac{n}{2}$ cycles.

For any $k < \frac{n}{4}$, all nodes in $G$ and $H$ have the isomorphic $k$-hop neighborhood, namely a path of length $2k$. Therefore, by Lemma C.2, all rows of the power of the adjacency matrices $A_{v:}^k$ have the same entries. Moreover, if we consider the concatenated tensors $(A, \dots, A^k)$, then the "row" $(A, \dots, A^k)_{:v:}$ will also have the same set of length-$k$ vectors. If follows that at each iteration, all nodes in $G$ and $H$ will have the same $k$-Walk-WL colors.

Conversely, $G$ and $H$ can be distinguished by one iteration of the 1-EP-WL test. The smallest Laplacian eigenvalue of $G$ is 0 with multiplicity 1 as $G$ is connected, and the smallest Laplacian eigenvalue of $H$ is 0 with multiplicty 2 as $H$ is disconnected. Moreover, the projection onto the zero eigenspace of $G$ is $\frac{1}{n}J$ where $J_n$ is the $n \times n$ all-ones matrix, while the projection onto the zero eigenspace of $H$ is $\frac{1}{n/2}\begin{bmatrix} J_{n/2} & 0 \\ 0 & J_{n/2} \end{bmatrix}$. 1-EP-WL can tell these matrices apart as $H$'s projection matrix has zero entries. □

## D. $k$-Harmonic Distances

### D.1. Preliminaries: Sparse-$\psi$-WL test

In this section, we present the *sparse-$\psi$-WL test*, a modification of the 1-WL test that incorporates edge features. While the WL test provides an upper bound on the expressive power of MPNNs, the sparse-$\psi$-WL test upper bounds MPNNs *that use edge features* decided by $\psi$. In the rest of this section, we present several results about the expressivity of this sparse-$\psi$-WL test when $\psi$ is a $k$-harmonic distance.

An ***edge positional encoding*** is a function $\psi$ that assigns each graph $G$ a map $\psi_G : E_G \to \mathbb{R}^m$ such that, if $\sigma : E_G \to E_H$ is the map on edges induced by a graph isomorphism, then $\psi_G = \psi_H \circ \sigma$. In the current work, we usually take $\psi$ to be a $k$-harmonic distance, i.e., $\psi((u,v)) = H^k(u,v) \in \mathbb{R}$.

Like the GD-WL test, the ***Sparse-$\psi$-WL-test*** iteratively assigns labels to the vertices of a graph. The labels for the $t^{\text{th}}$ iteration are computed using the following formula:

$$\chi_\psi^{(t)}(v) = \text{hash}\left( \chi_\psi^{(t-1)}(v), \left\{\!\!\left\{ \left( \psi((u,v)), \chi_\psi^{(t-1)}(u) \right) : (u,v) \in E \right\}\!\!\right\} \right)$$

### D.2. Proof of Theorem 4.4

**Theorem 4.4.** *Let $k \ge 1$. The Sparse-$k$-Harmonic-WL test is strictly stronger than the WL test.*

*Proof.* Let $C_n$ denote the ring graph on $n$ nodes. In particular, let $C_9$ be the ring graph on 9 nodes, and let $C_{3,3} = \cup_{i=1}^3 C_3$ be the graph that is the union of 3 ring graphs on 3 nodes; see Figure 2. Observe that $C_9$ and $C_{3,3}$ are indistinguishable by the WL test as they are both 2-regular graphs.

Now, we will show that $C_9$ and $C_{3,3}$ are distinguishable by one iteration of the Sparse-$k$-Harmonic-WL for any $k \ge 1$.

First, for any edges $e, e' \in C_9$, $H_{C_9}^k(e) = H_{C_9}^{k'}(e')$; likewise for any two edge in $C_{3,3}$. This is because both graphs are edge-transitive. Second, for any edge $e \in E_{C_{3,3}}$ and any edge $e' \in E_{C_3}$, $H_{C_{3,3}}^k(e) = H_{C_3}^{k'}(e')$. This is because for a graph

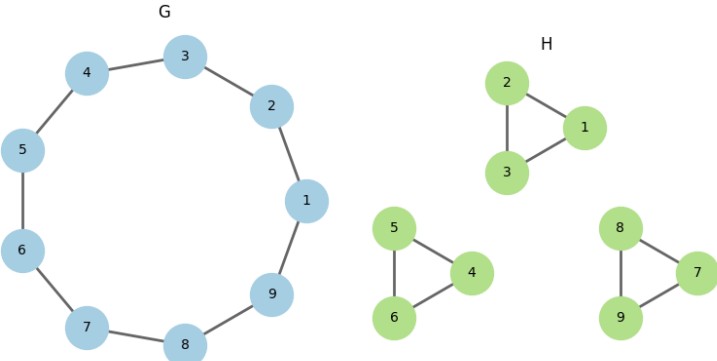

*Figure 2.* The graphs $G$ (the length 9 cycle graph) and $H$ (3 copies of the length 3 cycle graph) are indistinguishable by the WL test, but distinguishable by any Sparse-$k$-Harmonic-WL test

$G$ with connected components $G_1, \ldots, G_k$,

$$
L_G^{k+} = \begin{bmatrix} L_{G_1}^{k+} & 0 & \cdots \\ 0 & L_{G_2}^{k+} & \\ \vdots & & \ddots \end{bmatrix}
$$

This follows from the fact that the eigenvectors of a disconnected graph are the eigenvectors of each of its connected components with zero padding to be the correct dimensionality; see (Spielman, 2025, Lemma 3.1.1).

As $C_9$ is a regular graph and all edges have the same $k$-harmonic distance, then all nodes in $C_9$ have the same Sparse-$k$-Harmonic-WL color; likewise for $C_{3,3}$. For both graphs, these colors are $\chi^{(1)}(v) = (1, \{\!\{(1, H^k(e), (1, H^k(e)\}\!\})$, where $H^k(e)$ denotes the unqiue $k$-harmonic distance in the respective graph. Therefore, we only need to show that $H_{C_9}^k(e) \neq H_{C_3}^k(e')$ for $e \in E_{C_9}$ and $e' \in C_3$.

We first derive an exact formula for the $k$-harmonic distances of edges in these graphs.

**Lemma D.1.** *Let $C_{2n+1}$ be the cycle graph on $2n + 1$ vertices. Let $k > 0$. Then the $k$-harmonic distance of any edge in* $C_{2n+1}$ *is* $H_{C_{2n+1}}^k(e) = \frac{2}{2n+1} \sum_{t=1}^{n} \left(2 - 2\cos\left(\frac{2\pi t}{2n+1}\right)\right)^{-(k-1)}$

*Proof.* The analytical form of the eigenvectors and eigenvalues of cycle graphs are well-established (Spielman, 2025, p. 49). In addition to the all-ones vector with eigenvalue 0 (which is an eigenpair of the Laplacian of all graphs), for all $1 \leq t \leq n$, there are two distinct eigenvalues corresponding $x_t$ and $y_t$ corresponding to the eigenvalue $\lambda_t$. Let the vertices of $C_{2n+1}$ as the integers $\{0, ..., 2n\}$. The eigenvectors are

$$
\mathbf{x}_t(i) = \sqrt{\frac{2}{2n+1}} \cdot \cos\left(\frac{2\pi t i}{2n+1}\right), \quad \mathbf{y}_t(i) = \sqrt{\frac{2}{2n+1}} \cdot \sin\left(\frac{2\pi t i}{2n+1}\right)
$$

with eigenvalue

$$
\lambda_t = 2 - 2\cos\left(\frac{2\pi t}{2n+1}\right).
$$

Further, the $k^{\text{th}}$ power of the pseudoinverse of $G$ is

$$
(L^+)^k = \sum_{t=1}^{n} (\lambda_t)^{-k} \left(x_t x_t^T + y_t y_t^T\right)
$$

Thus, the $k$-harmonic distance of an edge in $C_{2n+1}$ is

$$
\begin{aligned}
H^k_{C_{2n+1}}(i, i+1) =& (1_i - 1_{i+1})^T L^{+k}(1_i - 1_{i+1}) \\
=& \sum_{t=1}^{n} \left(2 - 2\cos\left(\frac{2\pi t}{2n+1}\right)\right)^{-k} \frac{2}{2n+1}\left[\cos^2\left(\frac{2\pi ti}{2n+1}\right) + \sin^2\left(\frac{2\pi ti}{2n+1}\right)\right. \\
& - 2\sin\left(\frac{2\pi ti}{2n+1}\right)\sin\left(\frac{2\pi t(i+1)}{2n+1}\right) - 2\cos\left(\frac{2\pi ti}{2n+1}\right)\cos\left(\frac{2\pi t(i+1)}{2n+1}\right) \\
& \left.+ \sin^2\left(\frac{2\pi t(i+1)}{2n+1}\right) + \cos^2\left(\frac{2\pi t(i+1)}{2n+1}\right)\right]
\end{aligned}
$$

Here $0 \le i < 2n$. The choice of $i$ is arbitrary as all edges in $C_{2n+1}$ have the same $k$-harmonic distance. By applying the common trigonometry identities $\sin^2(x) + \cos^2(x) = 1$ and $\cos(x - y) = \cos(x)\cos(y) + \sin(x)\sin(y)$, we arrive at

$$
\begin{aligned}
H^k_{C_{2n+1}}(i, i+1) =& \frac{2}{2n+1}\sum_{t=1}^{n}\left(2 - 2\cos\left(\frac{2\pi t}{2n+1}\right)\right)^{-k}\left[2 - 2\cos\left(\frac{2\pi t}{2n+1}\right)\right] \\
=& \frac{2}{2n+1}\sum_{t=1}^{n}\left(2 - 2\cos\left(\frac{2\pi t}{2n+1}\right)\right)^{-(k-1)} \qquad\qquad \square
\end{aligned}
$$

Observe that for the case of the disconnected 3-cycle graph $(n = 1)$, the $k$-harmonic of an edge is

$$
H^k_{C_3}(e) = \frac{2}{3}\left(2 - 2\cos\left(\frac{2\pi}{3}\right)\right)^{-(k-1)}
$$

and for the case of the 9-cycle graph $(n = 4)$ we have

$$
\begin{aligned}
H^k_{C_9}(e) =& \frac{2}{9}\left(2 - 2\cos\left(\frac{2\pi}{9}\right)\right)^{-(k-1)} + \frac{2}{9}\left(2 - 2\cos\left(\frac{4\pi}{9}\right)\right)^{-(k-1)} \\
&+ \frac{2}{9}\left(2 - 2\cos\left(\frac{2\pi}{3}\right)\right)^{-(k-1)} + \frac{2}{9}\left(2 - 2\cos\left(\frac{8\pi}{9}\right)\right)^{-k-1}
\end{aligned}
$$

It is easy to see that

$$
H^k_{C_9}(e) = \frac{1}{3}H^k_{C_3}(e) + \frac{2}{9}\left[\left(2 - 2\cos\left(\frac{2\pi}{9}\right)\right)^{-k-1} + \left(2 - 2\cos\left(\frac{4\pi}{9}\right)\right)^{-k-1} + \left(2 - 2\cos\left(\frac{8\pi}{9}\right)\right)^{-k-1}\right]
$$

or more simply

$$
H^k_{C_9}(e) = \frac{1}{3}H^k_{C_3}(e) + f(k)
$$

so we want to show that $f(k) > \frac{2}{3}H^k_{C_3}$. Because $\cos(\pi x)$ is strictly decreasing on the interval $x \in [0, 1]$ this implies that

$$
0 < 2 - 2\cos\left(\frac{2\pi}{9}\right) < 2 - 2\cos\left(\frac{4\pi}{9}\right) < 2 - 2\cos\left(\frac{2\pi}{3}\right) < 2 - 2\cos\left(\frac{8\pi}{9}\right)
$$

Accordingly, for $k \ge 1$,

$$\frac{2}{9}\left(2 - 2\cos\left(\frac{2\pi}{9}\right)\right)^{-(k-1)} \geq \frac{2}{9}\left(2 - 2\cos\left(\frac{4\pi}{9}\right)\right)^{-(k-1)}$$

$$\geq \frac{2}{9}\left(2 - 2\cos\left(\frac{2\pi}{3}\right)\right)^{-(k-1)} = \frac{1}{3}H_{C_3}^k(e)$$

$$\geq \frac{2}{9}\left(2 - 2\cos\left(\frac{8\pi}{9}\right)\right)^{-(k-1)} > 0$$

The first two terms are larger than $\frac{1}{3}H_{C_3}^k(e)$, with the last term being strictly positive regardless. This implies that

$$f(k) > \frac{2}{3}H_{C_3}^k$$

or that $H_{C_9}^k(e)$ and $H_{C_3}^k(e)$ are never equal for any value of $k \geq 1$. This implies that any value of $k$ used for the sparse $k$-harmonic test will be able to successfully distinguish any two edges in $C_{3,3}$ and $C_9$. Therefore, a single iteration of Sparse-$k$-Harmonic-WL will result in different multisets for $G$ and $H$ as the edge features will be aggregated to their incident nodes. Thus, as we have found a pair of graphs that Sparse-$k$-Harmonic-WL can distinguish that 1-WL cannot, Sparse-$k$-Harmonic-WL > WL. $\qquad\square$

**Corollary D.1.** *Sparse-$k$-Harmonic-WL can distinguish any two odd cycle ring graphs of the form $C_n$ and $\frac{n}{m}$ copies of $C_m$ where $m|n$.*

*Proof.* Observe that the logic in the previous proof follows similarly when $n$ is varied up to

$$\sum_{t=1}^{n} \frac{2}{n}\left(2 - 2\cos\left(\frac{2\pi t}{2n+1}\right)\right)^{-k-1}$$

for the rest of the proof to hold, we need to deduce that $H_{C_m}^k(e) \subset H_{C_n}^k(e)$. For this to be true for two summations of the form $\cos(2\pi t/2n + 1)$ it must be the case that $m|n$. While it is true that the two summations will share terms when $m$ and $n$ are not coprime, for $H_{C_m}^k(e) \subset H_{C_n}^k(e)$ it must be that $m|n$.

From here, the rest of the proof remains true. $\qquad\square$

### D.3. Proof of Theorem 4.5

We use the following lemma about the number of roots of an exponential function. A stronger variant of it is proved by Jameson (2006, Theorem 3.1).

**Lemma D.2.** *Let $f(x) = \sum_{i=1}^{t} a_i b_i^x$, with nonzero $a_i$s and positive $b_i$s. Then, $f(x) = 0$ for at most $t$ values of $x$.*

**Theorem 4.5.** *The 3-WL test is strictly stronger than the $k$-Harmonic-WL test and Sparse-$k$-Harmonic-WL tests for all $k \in \mathbb{R}$.*

*Proof.* We rely on the important results from (Zhang et al., 2024), which proves that 3-WL upper-bounds the EP-WL. We will thus show that Sparse-$k$-Harmonic-WL is upper bounded by EP-WL. The proof that $k$-Harmonic-WL is upper bounded by EP-WL is analogous. Recall that EP-WL is defined as:

$$\chi_{\mathcal{P}}^{(t+1)}(v) = \left(\chi_{\mathcal{P}}^{(t)}(v), \{\!\!\{\chi_{\mathcal{P}}^{(t)}(u), \mathcal{P}(u,v) : u \in V\}\!\!\}\right) \tag{1}$$

where $\mathcal{P}^L(u,v)$ is the eigenspace projection invariant associated with graph Laplacian $L$. Specifically, the Laplacian can be defined as

$$L = \sum_{i \in m} \lambda_i P_i \tag{2}$$

where $\lambda_i$ are the distinct eigenvalues and $P_i$ are the projection matrices for $m$ unique eigenvalues $\lambda_i$. The eigenspace projection invariant is the multiset

$$\mathcal{P}(u,v) = \{\!\{(\lambda_1, P_1(u,v)), \ldots, (\lambda_m, P_m(u,v))\}\!\}.$$

The outline for the rest of the proof is as follows. We aim to upper bound Sparse-$k$-Harmonic-WL by EP-WL. In order to prove our upper bound, we need to prove that both 1.) EP-WL can determine the $k$-harmonic distance of a pair of nodes and 2.) EP-WL can successfully recover which pairs of nodes are connected by an edge. Part 1) is implied by Lemmas D.3 and D.5 (and proved in the proof of Lemma D.6) and part 2) is Corollary D.2.

We begin with a few observations about EP-WL.

**Lemma D.3.** *Let $G$ and $H$ be graphs. Let $u, v \in V_G$ and $x, y \in V_H$. Then $\mathcal{P}(u,v) = \mathcal{P}(x,y)$ if and only if $L_G^k(u,v) = L_H^k(x,y)$ for all $k \in \mathbb{R}$*

*Proof of Lemma D.3.* If $\mathcal{P}(v,v) = \mathcal{P}(u,w)$, then for any $k$,

$$L_G^k(u,v) = \sum_{i=1}^m \lambda_{G,i}^k P_{i,G}(u,v) = \sum_{i=1}^m \lambda_{H,i}^k P_{i,H}(x,y) = L_H^k(x,y).$$

Now assume $\mathcal{P}(v,v) \neq \mathcal{P}(u,w)$. Consider the polynomial

$$L_G^k(u,v) - L_H^k(x,y) = \sum_{i=1}^m \lambda_{G,i}^k P_{G,i}(u,v) - \sum_{i=1}^m \lambda_{H,i}^k P_{H,i}(x,y)$$

As $\mathcal{P}(v,v) \neq \mathcal{P}(u,w)$, then there is some $i$ such that $\lambda_{i,G} \neq \lambda_{i,H}$ or $P_{G,i}(u,v) \neq P_{H,i}(x,y)$. In either case, this polynomial is not the zero polynomial. Thus, by Lemma D.2, there must be some $k$ such that $L_G^k(u,v) - L_H^k(x,y) \neq 0$, and so $L_G^k(u,v) \neq L_H^k(x,y)$. $\qquad\square$

**Corollary D.2.** *Let $G$ and $H$ be graphs. Let $u, v \in V_G$ and $x, y \in V_H$. If $\mathcal{P}(u,v) = \mathcal{P}(x,y)$, then $(u,v) \in E_G$ if and only if $(x,y) \in E_H$*

*Proof of Corollary D.2.* $L_G(u,v) < 0$ if and only if $(u,v) \in E_G$, so this follows from Lemma D.3. $\qquad\square$

In what follows, an ***isolated vertex*** is a vertex with 0 neighbors.

**Corollary D.3.** *Let $G$ and $H$ be graph. Let $v \in V_G$ and $u, w \in V_H$. If $v$ is not an isolated vertex, then $\mathcal{P}(v,v) = \mathcal{P}(u,w)$ only if $u = w$.*

*Proof of Corollary D.3.* $L_H(u,w) > 0$ only if $u = w$ and $u$ is not an isolated vertex, so this follows from Lemma D.3. $\quad\square$

**Lemma D.4.** *Let $G$ and $H$ be graph. Let $v \in V_G$ and $x \in V_H$. If $\chi_{\mathcal{P}}^{(1)}(v) = \chi_{\mathcal{P}}^{(1)}(x)$, then either both $v$ and $x$ are isolated vertices or neither $v$ and $x$ are isolated vertices.*

*Proof of Lemma D.4.* If $v$ is an isolated vertex, for any $u \in V$, $L_G(u,v) = 0$. Therefore, as $\{\!\{\mathcal{P}(u,v) : v \in V_G\}\!\} = \{\!\{\mathcal{P}(x,y) : y \in V_H\}\!\}$, by Lemma D.3, it must also be the case that $L_H(x,y) = 0$ for all $y \in V_H$. $\qquad\square$

**Lemma D.5.** *Let $G$ and $H$ be graph. Let $v \in V_G$ and $x \in V_H$. If neither $v$ and $x$ are isolated vertices and $\chi_{\mathcal{P}}^{(1)}(v) = \chi_{\mathcal{P}}^{(1)}(x)$, then $L^k(v,v) = L^k(x,x)$ for all $k \in \mathbb{R}$.*

*Proof of Lemma D.5.* If $\chi_{\mathcal{P}}^{(1)}(u) = \chi_{\mathcal{P}}^{(1)}(v)$, then this implies that $\{\!\{\mathcal{P}(u,v) : v \in V_G\}\!\} = \{\!\{\mathcal{P}(x,y) : v \in V_G\}\!\}$. As $v$ and $x$ are not isolated, then by Corollary D.3, $\mathcal{P}(v,v) = \mathcal{P}(x,x)$. Thus, Lemma D.3 implies the lemma. $\qquad\square$

Recall that the $k$-harmonic distance is

$$H^k(s,t) = \sqrt{L^{+k}(s,s) + L^{+k}(t,t) - 2L^{+k}(s,t)}$$

Let $\chi_k^{(t)}(v)$ denote the Sparse-$k$-Harmonic-WL color.

**Lemma D.6.** *Let $G$ and $H$ be graphs. Let $v \in V_G$ and $x \in V_H$. For all $t \geq 0$, if $\chi_{\mathcal{P}}^{(t+1)}(v) = \chi_{\mathcal{P}}^{(t+1)}(x)$, then $\chi_k^{(t)}(v) = \chi_k^{(t)}(x)$.*

*Proof of Lemma D.6.* We prove this by induction on $t$. For $t = 0$, this is trivial as all vertices have the same Sparse-$k$-Harmonic-WL color.

Now assume this is true for some $t - 1$. We will prove it is the case for $t$.

If $\chi_{\mathcal{P}}^{(t+1)}(v) = \chi_{\mathcal{P}}^{(t+1)}(x)$, then by Lemma D.4, there are two cases: both $v$ and $x$ are isolated vertices or neither are.

If $v$ and $x$ are isolated vertices, then $\chi_k^{(t)}(v) = \chi_k^{(t)}(x)$ as all isolated vertices have the Sparse-$k$-Harmonic-WL color.

If $v$ and $x$ are not isolated vertices, then

$$\left(\chi_{\mathcal{P}}^{(t)}(v), \{\!\!\{\chi(u), \mathcal{P}(u,v) : u \in V_G\}\!\!\}\right) = \left(\chi_{\mathcal{P}}^{(t)}(x), \{\!\!\{\chi(y), \mathcal{P}(x,y) : y \in V_H\}\!\!\}\right).$$

By the induction hypothesis, the first part of the tuple implies that $\chi_k^{(t-1)}(v) = \chi_k^{(t-1)}(x)$.

Next, observe that by Corollary D.2 that

$$\{\!\!\{\chi_{\mathcal{P}}^{(t)}(u), \mathcal{P}(u,v) : u \in V_G\}\!\!\} = \{\!\!\{\chi_{\mathcal{P}}^{(t)}(y), \mathcal{P}(x,y) : (x,y) \in V_H\}\!\!\}$$

$$\Rightarrow \{\!\!\{\chi_{\mathcal{P}}^{(t)}(u), \mathcal{P}(u,v) : (u,v) \in E_G\}\!\!\} = \{\!\!\{\chi_{\mathcal{P}}^{(t)}(y), \mathcal{P}(x,y) : (x,y) \in E_H\}\!\!\}$$

Thus, there is a bijection $\sigma : N(v) \to N(X)$ such that $(\chi_{\mathcal{P}}^{(t)}(u), \mathcal{P}(u,v)) = (\chi_{\mathcal{P}}^{(t)}(\sigma(u)), \mathcal{P}(x,\sigma(u)))$ for all $u \in N(v)$. We claim that for each $u \in N(v)$ that $(\chi_{k-1}^{(t)}(u), H^k(u,v)) = (\chi_k^{(t-1)}(\sigma(u)), H^k(x,\sigma(u)))$. As $\chi_{\mathcal{P}}^{(t)}(u) = \chi_{\mathcal{P}}^{(t)}(\sigma(u))$, the inductive hypothesis implies that $\chi_{k-1}^{(t)}(u) = \chi_k^{(t-1)}(\sigma(u))$. To prove that $H^k(u,v) = H^k(x,\sigma(u))$, first observe that because $v$ and $x$ are not isolated vertices and $\chi_{\mathcal{P}}^{(t+1)}(v) = \chi_{\mathcal{P}}^{(t+1)}(x)$, then $L_G^{+k}(v,v) = L_G^{+k}(x,x)$ by Lemma D.5. Likewise, $L_G^{+k}(u,u) = L_G^{+k}(\sigma(u),\sigma(u))$. Finally, as $\mathcal{P}(u,v) = \mathcal{P}(x,\sigma(u))$, then $L^{+k}(u,v) = L^{+k}(x,\sigma(u))$ by Lemma D.3. Therefore, $H^k(u,v) = H^k(x,\sigma(u))$. As we have shown there is a bijection $\sigma : N(v) \to N(X)$ such that $(\chi_{k-1}^{(t)}(u), H^k(u,v)) = (\chi_k^{(t-1)}(\sigma(u)), H^k(x,\sigma(u)))$ for all $u \in N(v)$, this concludes our proof that $\chi_k^{(t)}(v) = \chi_k^{(t)}(x)$ □

We can now use this lemma to prove the theorem. If $G$ and $H$ are 3-WL indistinguishable, they are EP-WL indistinguishable by Zhang et al. (2024). If $G$ and $H$ are EP-WL indistinguishable, this implies that $\{\!\!\{\chi_{\mathcal{P}}^{(t)}(v) : v \in V_G\}\!\!\} = \{\!\!\{\chi_{\mathcal{P}}^{(t)}(x) : x \in V_H\}\!\!\}$ for all $t \geq 0$. Lemma D.6 then implies $\{\!\!\{\chi_k^{(t)}(v) : v \in V_G\}\!\!\} = \{\!\!\{\chi_k^{(t)}(x) : x \in V_H\}\!\!\}$, so $G$ and $H$ are Sparse-$k$-Harmonic-WL indistinguishable. □

### D.4. Proof of Theorem 4.7

The **$k$-hop neighbor** of radius $k$ around a node $v$ is the graph $(B_k(v), E_k(v))$ with nodes $B_k(v) = \{u \in V : d(v,u) \leq k\}$ and edges $E_k(v) = \{\{u,w\} : d(v,u) \leq k-1, d(v,w) \leq k\}$. Two nodes $u$ and $v$ have **isomorphic $k$-hop neighborhoods** if there is a graph isomorphism $\sigma : B_k(u) \to B_k(v)$ such that $\sigma(u) = v$. While the following lemma is folklore, we will prove a stronger version of this theorem in the coming section (proof of Lemma D.9), so readers interested in a proof of this lemma are encouraged to read that proof.

**Lemma D.7** (Folklore). *Let $G$ and $H$ be graphs, and let $v \in V_G$ and $u \in V_H$. If $v$ and $u$ have isomorphic $k$-hop neighborhoods, then the WL colors $\chi^{(l)}(v) = \chi^{(l)}(x)$ for all $0 \leq l \leq k$.*

**Lemma D.8** ((Black et al., 2024a, Theorem 5.1)). *Let $G = (V, E)$ be a connected graph. Let $(u, v) \in E$ be a cut edge, and let $S, T \subset V$ be the connected components of $G$ after removing the edge $(u, v)$. Then*

$$B(u, v)^2 = \frac{|S||T|}{|V|}.$$

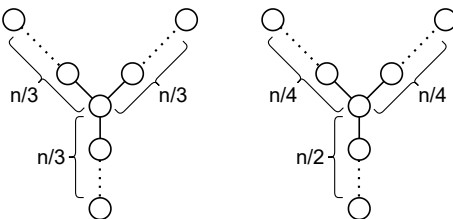

*Figure 3.* Two non-isomorphic trees that Sparse-Biharmonic-WL can distinguish in 1 iteration but Sparse-Resistance-WL cannot distinguish in $o(n)$ iterations.

*Proof of Theorem 4.7.* Let $n$ be any positive integer that is divisible by 12. We consider the pair of rooted trees $G$ and $H$ where $G$ is a root connected to three paths of length $n/3$ and $H$ is a root connected to two paths of length $n/4$ and one path of length $n/2$; see Figure 3 for a picture of $G$ and $H$. Observe that these graphs are both trees and both have $n + 1$ vertices. We will show that $G$ and $H$ are indistinguishable by $\lfloor n/8 \rfloor$ iterations of the WL test, but are distinguishable by a single iteration of the Sparse-Biharmonic-WL test.

First, we show that these graphs are distinguishable by one iteration of Sparse-Biharmonic-WL. First, observe that because $G$ and $H$ are both trees, then all edges in either graph is a cut edge; accordingly, we can use Lemma D.8 to compute the biharmonic distance of all edges in each graph. In particular, consider the edge $e$ connecting the root of $H$ to the path of length $n/2$. The squared biharmonic distance of $e$ is $B(e)^2 = \frac{(n/2)(n/2+1)}{n+1}$; any edge $e'$ in $G$ has biharmonic distance at most $B(e')^2 \le \frac{(n/3)(2n/3+1)}{n+1} < \frac{(n/2)(n/2+1)}{n+1} = B(e)^2$. Therefore, the Sparse-Biharmonic-WL color of the root $\chi_B^{(1)}(r_H)$ of $H$ contains an edge with biharmonic distance $B(e) = \sqrt{\frac{(n/2)(n/2+1)}{n+1}}$. As there is no edge in $G$ with biharmonic distance $\sqrt{\frac{(n/2)(n/2+1)}{n+1}}$, there is no node in $G$ with the same Sparse-Biharmonic-WL color as $r_H$. Therefore, one iteration of Sparse-Biharmonic-WL distinguishes $G$ and $H$.

Next, we need to show that $G$ and $H$ cannot be distinguished in $\lfloor \frac{n}{8} \rfloor$ iterations of the WL test. First, we observe that for any $k < \lfloor \frac{n}{8} \rfloor$, the $k$-hop neighborhoods of the nodes in $G$ and $H$ are of one of three types:

1. Nodes that are distance $r < k$ from a leaf of a tree The $k$-hop neighborhoods of these nodes are the node connected to a path of length $r$ (the path connecting the node to the leaf) and a path of length $k$.

2. Nodes that are distance $r < k$ from the root The $k$-hop neighborhoods of these nodes are the node connected to a path of length $k$ and a path of length $r$ connected to two paths of length $k - r$.

3. Nodes that are distance $> k$ from both a leaf and the root The $k$-hop neighborhood of these nodes are the node connected to two paths of length $k$.

As $k < \lfloor \frac{n}{8} \rfloor$, there are no nodes of that are both distance $< k$ to a leaf and a root, as the distance between any leaf and a root in either tree is $\frac{n}{4}$

For any $0 < r < k$, in both graphs, there are three nodes of distance exactly $r$ to a leaf and distance exactly $r$ to the root. For $r = 0$, there is one node of distance $r$ to the root (the root itself) and three nodes of distance $r = 0$ to the leaves (the leaves themselves.) The remaining $n - 6k$ nodes of the graph are at distance $> k$ from both a leaf to a root. As there is a bijection from the nodes of $G$ to the nodes of $H$ such that paired nodes have isomorphic $k$-hop neighborhoods, then by Lemma D.7, we conclude that $G$ and $H$ are indistinguishable by $k$ iterations of the WL test for all $k < \lfloor \frac{n}{8} \rfloor$. □

## D.5. Example of Theorem 4.7

In Theorem 4.7, we asserted that there are trees that Sparse-Biharmonic-WL can distinguish in one iteration. However, this does not generalize to all non-isomorphic trees. We provide one such counterexample in Figure 4 where it would take both Sparse-Biharmonic-WL and 1-WL $\Omega(n)$ iterations to distinguish these two trees.

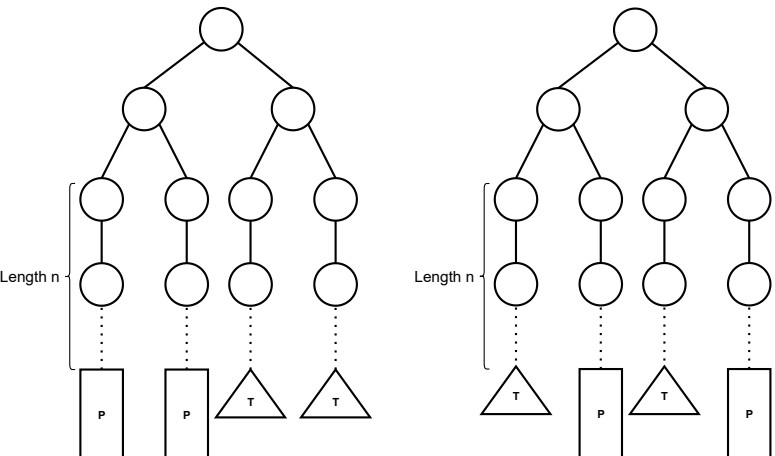

*Figure 4.* Two non-isomorphic trees $G$ and $H$ with $n$ vertices that Sparse-Biharmonic-WL takes $o(n)$ iterations to distinguish. Let $T$ be the complete tree consisting of $n$ nodes and $P$ be a path of $n$ nodes.

**Theorem D.1.** *There exist pairs of graph $G$ and $H$ with $n$ nodes that cannot be distinguished in $o(n)$ iterations of the Sparse-Biharmonic-WL test.*

To prove this, we will use a variant of Lemma D.8 for the sparse $\psi$ WL test. For an edge positional encoding $\psi$ and two graphs $G$ and $H$, we define a **$\psi$-preserving isomorphism** as an isomorphism $\sigma : V_G \to V_H$ such that for each edge $(u, v) \in E_G$, $\psi(u, v) = \psi(\sigma(u), \sigma(v))$. In the following lemma, when we say a $\psi$-preserving isomorphism between neighborhoods, we define $\psi$ with respect to the entire graphs $G$ and $H$, and not with respect to the neighborhoods.

**Lemma D.9.** *Let $G$ and $H$ be graphs, and let $v \in V_G$ and $u \in V_H$. Let $\psi$ be an edge positional encoding. If there is a $\psi$-preserving isomorphism between the $k$-hop neighborhoods of $v$ and $x$, then the WL colors $\chi_\psi^{(l)}(v) = \chi_\psi^{(l)}(x)$ for all $0 \leq l \leq k$.*

*Proof.* We will actually prove a stronger result. If there is a $\psi$-preserving isomorphism $\sigma$ between the $k$-hop neighborhoods of $u$ and $v$, then for all $0 \leq l \leq k$ and for all vertices $u$ that are at most $k - l$ hops away from $v$, then $\chi_\psi^{(l)}(v) = \chi_\psi^{(l)}(\sigma(u))$. We will prove this by induction on $l$. As $u$ is 0 hops from itself, then this implies the theorem.

For the base case of $l = 0$, this is true by the definition of the sparse $\psi$ WL test.

Now assume this is true for some $l \geq 0$; we will prove it is true for $l + 1$. Consider a vertex $u$ that is at distance at most $k - (l + 1)$ from $v$. We claim that $\chi_\psi^{(l+1)}(u) = \chi_\psi^{(l+1)}(\sigma(u))$. The color of $v$ is defined

$$\chi_\psi^{(l+1)}(u) = (\chi_\psi^{(l)}(u), \{\!\!\{ (\chi_\psi^{(l)}(u), \psi(u, w)) : (u, w) \in E_G \}\!\!\}).$$

By the inductive hypothesis, we know that $\chi_\psi^{(l)}(u) = \chi_\psi^{(l)}(\sigma(u))$ as $u$ is at most distance $k - (l + 1) < k - l$ from $v$.

Moreover, as $\sigma$ is an isomorphism, then the neighbors of $u$ are $\{\sigma(w) : (u, w) \in E_G\} = \{y : (\sigma(u), y) \in E_H\}$. Moreover, any neighbor of $u$ is at most distance $k - l$ from $v$, so $\chi_\psi^{(l)}(w) = \chi_\psi^{(l)}(\sigma(w))$. Finally, as $\sigma$ is $\psi$-preserving, we know that

$\psi(u, w) = \psi(\sigma(u), \sigma(w))$. Therefore, we conclude that

$$
\begin{aligned}
\chi_\psi^{(l+1)}(u) &= (\chi_\psi^{(l)}(u), \{\!\!\{(\chi_\psi^{(l)}(u), \psi(u, w)) : (u, w) \in E_G\}\!\!\}) \\
&= (\chi_\psi^{(l)}(\sigma(u)), \{\!\!\{(\chi_\psi^{(l)}(\sigma(u)), \psi(\sigma(u), \sigma(w))) : (u, w) \in E_G\}\!\!\}) \\
&= (\chi_\psi^{(l)}(\sigma(u)), \{\!\!\{(\chi_\psi^{(l)}(\sigma(u)), \psi(\sigma(u), y)) : (\sigma(u), y) \in E_H\}\!\!\}) \\
&= \chi_\psi^{(l+1)}(\sigma(u)) \qquad\qquad\qquad\qquad\qquad\qquad\qquad\qquad\qquad\qquad \square
\end{aligned}
$$

### D.6. Proof of Theorem 4.8

**Theorem D.2.** *Let $G$ and $H$ be two graphs such that the Laplacians $L_G$ and $L_H$ have $t_G$ and $t_H$ distinct non-zero eigenvalues respectively. Let $u, v \in V_G$ and $x, y \in V_H$. Then, either*

*(1) the $k$-harmonic distances $H^k(u, v) = H^k(x, y)$ for all $k \in \mathbb{R}$, or*

*(2) the $k$-harmonic distances $H^k(u, v) = H^k(x, y)$ for at most $t_G + t_H$ values of $k \in \mathbb{R}$.*

*Proof.* Suppose that the Laplacian of $G$ has $t_G$ distinct non-zero eigenvalues. Let $L_G$ be the Laplacian of $G$, let $0 < \lambda_1 < \lambda_2 < \cdots < \lambda_t$ be its nonzero distinct Laplacian eigenvalues, let $U_i$ be the matrix with columns that are an orthogonal basis for the eigenspace of $U_i$. Then the Laplacian of $G$ is

$$
L_G = \sum_{i=1}^{t} \lambda_i U_i U_i^T.
$$

Therefore, the $k$-harmonic distance between two vertices $u$ and $u$ is

$$
\begin{aligned}
(1_u - 1_v)^T (L_G^+)^k (1_u - 1_v) &= (1_u - 1_v)^T \left( \sum_{i=2}^{n} \lambda_i^{-k} U_i U_i^T \right)(1_u - 1_v) \\
&= \sum_{i=1}^{t} \lambda_i^{-k} (U_i^T(1_u - 1_v))^T U_i^T(1_u - 1_v) \\
&= \sum_{i=1}^{t} \lambda_i^{-k} p_i^2(u, v),
\end{aligned}
$$

where $p_i(x, y) = \|U_i^T(1_x - 1_y)\|_2$.

We can similarly write $L_H$ as the decomposition of its eigenvalues and eigenvectors. To distinguish the eigenvectors and eigenvalues of $G$ and $H$, we will denote each with a subscript $G$ and $H$

It follows that if $G$ and $H$ have the same number $t$ of distinct eigenvalues, these eigenvalues are equal ($\lambda_{i,G} = \lambda_{i,H}$ for $1 \leq i \leq t$), and $p_i(u, v) = p_i(x, y)$ for all $1 \leq i \leq t$, then these pairs must have the same $k$-harmonic distance for all values of $k$.

Otherwise, either $G$ and $H$ have a different number of distinct eigenvalues, there is an eigenvalue $\lambda_{G,i} \neq \lambda_{H,i}$, or there exists at least one $2 \leq i \leq t$ for which $p_i(x, y) \neq p_i(u, v)$. Now, consider

$$
f(k) = (1_x - 1_y)^T(L_H^+)^k(1_x - 1_y) - (1_u - 1_v)^T(L_G^+)^k(1_u - 1_v) = \sum_{i=1}^{t_G} \lambda_{i,G}^{-k} p_i^2(u, v) - \sum_{i=1}^{t_H} \lambda_{i,H}^{-k} p_i^2(x, y)
$$

as a function $k \in \mathbb{R}$, i.e. $f : \mathbb{R} \to \mathbb{R}$ and $k$ is its only variable. Since this is an exponential function that is not identically zero, it has at most $t_G + t_H$ roots by Lemma D.2. The $k$-harmonic distances between $x, y$ and $u, v$ are different for all other values of $k$. $\qquad\square$

**Lemma D.10.** *Let $G$ and $H$ be graphs. Let $\psi$ and $\psi'$ be edge positional encodings such that, for all $u, v \in V_G$ and $x, y \in V_H$, $\psi(u, v) \neq \psi(x, y)$ implies $\psi'(u, v) \neq \psi'(x, y)$. Then if sparse-$\psi$-WL distinguishes $G$ and $H$, then sparse-$\psi'$-WL distinguishes $G$ and $H$.*

*Proof.* Let $\chi_\psi^{(t)}(v)$ denote the color of node $v$ under the sparse $\psi$ WL test at step $t$. Further, let $v_i \in G$ and $v_j \in H$. As a first step towards proving this lemma, we will show that if $\chi_\psi^{(t)}(v_i) \neq \chi_\psi^{(t)}(v_j)$, then $\chi_{\psi'}^{(t)}(v_i) \neq \chi_{\psi'}^{(t)}(v_j)$, or that if nodes $v_i$ and $v_j$ are not the same color under the sparse $\psi$ WL test, they will not be the same color under the sparse $\psi'$ WL test. We will prove this by induction on $t$.

Base Case: For $t = 0$, this is vacuously true as $\chi_\psi^{(0)}(v_i) = \chi_\psi^{(0)}(v_j) = 1$ for all $v_i \in G$ and $v_j \in H$.

Induction Hypothesis: Suppose this is true for $t - 1 \geq 0$. That is, if $\chi_\psi^{(t-1)}(v_i) \neq \chi_\psi^{(t-1)}(v_j)$, then $\chi_{\psi'}^{(t-1)}(v_i) \neq \chi_{\psi'}^{(t-1)}(v_j)$

We will now prove this is true for $t$. Suppose $\chi_\psi^{(t)}(v_i) \neq \chi_\psi^{(t)}(v_j)$. By definition of the WL test, it is either the case that: the colors were different in the previous iteration of the test: $\chi_\psi^{(t-1)}(v_i) \neq \chi_\psi^{(t-1)}(v_j)$, or the nodes aggregated distinguishing information in step $t$: $\{\!\{\psi(v_i, x), \chi_\psi^{(t-1)}(x) : (v_i, x) \in E_G\}\!\} \neq \{\!\{\psi(v_j, y), \chi_\psi^{(t-1)}(y) : (v_j, y) \in E_H\}\!\}$

- Case 1: $\chi_\psi^{(t-1)}(v_i) \neq \chi_\psi^{(t-1)}(v_j)$ By the induction hypothesis, $\chi_{\psi'}^{(t-1)}(v_i) \neq \chi_{\psi'}^{(t-1)}(v_j)$. This implies that $\chi_{\psi'}^{(t)}(v_i) \neq \chi_{\psi'}^{(t)}(v_j)$

- Case 2: $\{\!\{\psi(v_i, x), \chi_\psi^{(t-1)}(x) : (v_i, x) \in E_G\}\!\} \neq \{\!\{\psi(v_j, y), \chi_\psi^{(t-1)}(y) : (v_j, y) \in E_H\}\!\}$ WLOG suppose that $v_i$ and $v_j$ have the same number of neighbors, as their multisets will be vacuously different if they don't. Given that they have the same number of neighbors but have different multisets of colors, we can conclude that for any bijection $\sigma : N(v_i) \to N(v_j)$, there is a vertex $u \in N(v_i)$ such that $(\psi(v_i, u), \chi_\psi^{(t-1)}(u)) \neq (\psi(v_j, \sigma(u)), \chi_\psi^{(t-1)}(\sigma(u)))$.

  If $\chi_\psi^{(t-1)}(u) \neq \chi_\psi^{(t-1)}(\sigma(u))$ then the induction hypothesis holds and $\chi_{\psi'}^{(t-1)}(u) \neq \chi_{\psi'}^{(t-1)}(\sigma(u))$. If $\psi(v_i, u) \neq \psi(v_j, \sigma(u))$, then we invoke our assumption to say $\psi'(v_i, u) \neq \psi'(v_j, u)$ and the statement holds.

Thus, in both cases, $\chi_{\psi'}^{(t)}(v_i) \neq \chi_{\psi'}^{(t)}(v_j)$.

To finish the proof, we show that $G$ and $H$ are distinguishable by the Sparse-$\psi'$-WL test. Given that $G$ and $H$ are distinguishable by sparse $\psi$ WL, there is some $t > 0$ such that $\{\!\{\chi_\psi^{(t)}(v_i) : v_i \in V_G\}\!\} \neq \{\!\{\chi_\psi^{(t)}(v_j) : v_j \in V_H\}\!\}$. So, for any bijection $\sigma : V_G \to V_H$, there is a vertex $v \in V_G$ such that $\chi_\psi^{(t)}(v) \neq \chi_\psi(\sigma(v))$. From the above, this implies that $\chi_{\psi'}^{(t)}(v) \neq \chi_{\psi'}(\sigma(v))$. Therefore, $\{\!\{\chi_{\psi'}^{(t)}(v_i) : v_i \in V_G\}\!\} \neq \{\!\{\chi_{\psi'}^{(t)}(v_j) : v_j \in V_H\}\!\}$, so sparse $\psi'$ WL also distinguishes $G$ and $H$. $\square$

An analogous theorem holds for the GD-WL test.

**Lemma D.11.** *Let $G$ and $H$ be graphs. Let $\psi$ and $\psi'$ be relative positional encodings such that, for all $u, v \in V_G$ and $x, y \in V_H$, $\psi(u, v) \neq \psi(x, y)$ implies $\psi'(u, v) \neq \psi'(x, y)$. Then if $\psi$-WL distinguishes $G$ and $H$, then $\psi'$-WL distinguishes $G$ and $H$.*

*Proof.* The proof of this is analogous to the proof of Lemma D.10 $\square$

**Theorem 4.8.** *Let $G$ and $H$ be graphs with $n$ vertices that are distinguishable by $k$-Harmonic-WL or Sparse-$k$-Harmonic-WL for some $k$. Then for all but $O(n^5)$ values of $k' \in \mathbb{R}^+$, $G$ and $H$ are distinguishable by the $k'$-harmonic-WL test or test Sparse-$k'$-harmonic WL test respectively.*

*Proof.* Let $G$ and $H$ be graphs that are distinguishable by $k$-harmonic WL. For all pairs of nodes $u, v \in V_G$ and $x, y \in v_H$, let

$$K(u, v, x, y) = \begin{cases} \emptyset & \text{if } H^k(u, v) = H^k(x, y) \text{ for all } k \in \mathbb{R} \\ \{k \in \mathbb{R} : H^k(u, v) = H^k(x, y)\} & \text{otherwise} \end{cases}$$

Now let $K = \cup_{u, v \in V_G, \, x, y \in V_H} K(u, v, x, y)$ and let $k' \in \mathbb{R} \setminus K$. By construction, for any pairs $u, v \in V_G$ and $x, y \in V_H$, if $H^k(u, v) \neq H^k(x, y)$, then $H^{k'}(u, v) \neq H^{k'}(x, y)$. Therefore, the $k'$-harmonic distance satisfies the conditions of Lemma D.11, so $G$ and $H$ are distinguishable by the $k'$-Harmonic WL test. Moreover, $G$ and $H$ each have at most $n - 1$ distinct eigenvalues, so the size of $K$ is $O(n^5)$.

A similar argument using Lemma D.10 can be used to prove the results for the sparse-$k$-harmonic WL test. $\square$

### D.7. Proof of Theorem 4.6

**Theorem 4.6.** *Let* $[2n] = \{1, \ldots, 2n\}$. *The* $[2n]$*-harmonic WL test is as strong as the EP-WL test*

*Proof.* We will prove the $[2n]$-harmonic WL test is as strong as the EP-WL test. By Theorem D.2, it is only the case that $H_G^k(u, v) = H_H^k(x, y)$ for $k \in [2n]$ if $H_G^k(u, v) = H_H^k(x, y)$ for all $k \in \mathbb{R}$. However, by the proof of Theorem D.2, it is only the case that $H_G^k(u, v) = H_H^k(x, y)$ for all $k \in \mathbb{R}$ if $G$ and $H$ have the same number $t$ of distinct eigenvalues, $\lambda_{i,G} = \lambda_{i,H}$ for $1 \leq i \leq t$, and $p_i(u, v) = p_i(x, y)$ for all $1 \leq i \leq t$. However, if all three of these conditions are true, then the eigenspace projection invariant $\mathcal{P}_G(u, v) = \{(\lambda_{i,G}, \Pi_{i,G}(u, v)) : 1 \leq i \leq l\} = \{(\lambda_{i,H}, \Pi_{i,H}(u, v)) : 1 \leq i \leq l\} = \mathcal{P}_H(x, y)$. Therefore, by Theorem D.2, the $[2n]$-harmonic WL is as strong as the EP-WL test. $\square$

### D.8. Runtime to Compute $k$-Harmonic Distance

The naive algorithm to compute the $k$-harmonic distances between all pairs of nodes take $O(n^3)$ time and goes roughly as follows: compute an eigendecompositon of $L$, take the eigenvalues to the negative $k$th power, add the eigenvalues and eigenspace projections to compute $L^{-k}$, then compute all pairs $k$-harmonic distance.

However, there is a faster algorithm to approximately compute all pairs $k$-harmonic distance for fixed $k$. This is a variant of the approximation algorithm to compute effective resistance of Spielman & Srivastava (2011); we outline the idea here but refer to this paper for a more careful presentation and analysis. The key observation is that the $k$-harmonic distance between a pair of nodes $u$ and $v$ is the Euclidean distance between the vectors $L^{-k/2}1_u$ and $L^{-k/2}1_u$, and we can approximate this distance by computing the distance between the vectors $\Pi L^{k/2}1_u$ and $\Pi L^{-k/2}1_u$ where $P \in \mathbb{R}^{O(\log n) \times n}$ is a Johnson-Lindenstrauss random projection matrix (Johnson & Lindenstrauss, 1984). The speed-up comes from the fact that to compute the matrix $\Pi L^{-k/2}$, we can solve $O(\log n)$ linear systems in the matrix $L^{k/2}$ (i.e., solve for the rows of $\Pi$.) We can approximately solve linear systems in $L$ in $O(m \, \text{poly} \log n)$ time (Spielman & Teng, 2004; Jambulapati & Sidford, 2025), so we can solve linear system in $L^{k/2}$ in $O(mk \, \text{poly} \log n)$ by performing $k/2$ solves on $L$. (For $k$ odd, we can instead find the vectors $\Pi B L^{-\lceil k/2 \rceil}$, where $B$ is the signed incidence matrix, as $B^T B = L$. Matrix-vector multiplication for this matrix only takes $O(m)$ time as this matrix has $2m$ non-zero entries.) In total, we can approximately compute the all pairs $k$-harmonic distance in $O(mk \, \text{poly} \log n + n^2 \log n)$ time and the $k$-harmonic distance on the edges in $O(mk \, \text{poly} \log n + m \log n) = O(mk \, \text{poly} \log n)$ time.

## E. Experiments

### E.1. BREC

The BREC dataset includes several families of graphs ranging from 1-WL indistinguishable, to 4-WL indistinguishable. We provide a quick overview of the dataset and justification for why the results we received are consistent with our theoretical results.

**Basic:** Consists of 60 pairs of 1-WL indistinguishable graphs.

**Regular:** Consists of 140 pairs of regular graphs, subdivided into different families of regular graph. 50 pairs of simple regular graphs which are 1-WL indistinguishable, 50 pairs of strongly regular graphs which are 3-WL indistinguishable, 20 pairs of 4-vertex condition graphs which are *at least* 3-WL indistinguishable, and 20 pairs of distance regular graphs which are *at least* 3-WL indistinguishable.

**Extension:** Consists of 100 pairs of graphs that sit between 1-WL indistinguishable and 3-WL distinguishable. These graphs were generated outside of the context of the WL hierarchy with methods such as substructure counting, node marking, and $n$-hop subgraphs. The authors claim that these graphs are meant to provide more granularity to the space between 1-WL and 3-WL.

**CFI:** Consists of 100 pairs of graphs generated by the intentionally difficult Cai, Furer, and Immerman method. 60 pairs of these graphs are 1-WL indistinguishable, 20 pairs are 3-WL indistinguishable, and a further 20 pairs are 4-WL indistinguishable.

Per Table 1, we see that our results generally line up with the distinguishability of these graphs (a realized expressivity

between 1-WL and 3-WL). Since the $k$-harmonics readily attain their maximal expressive power on BREC, we push further: rather than a transformer with dense $n^2$ PEs, we evaluate an MPNN with varying message-passing depths to (i) validate Theorem 4.4 and (ii) probe how little PE signaling a GNN needs to match the theory. In this setting, PEs scale with $|E|$ and are aggregated locally (neighbor messages) rather than via global attention; results are shown in Table 3.

In practice, a **single** message-passing layer suffices to separate most 1-WL indistinguishable pairs in BREC. The largest gap to the transformer appears on the CFI graphs, as expected, yet the $k$-harmonics remain highly competitive — even with smaller parameter budget and purely local aggregation.

*Table 3.* % Accuracy for each family of graph in BREC broken down by number of message passing layers for both Effective Resistance and Biharmonic Distance

| | **Resistance** | | | | **Biharmonic** | | | |
|---|---|---|---|---|---|---|---|---|
| **Layers** | 1 | 2 | 3 | 4 | 1 | 2 | 3 | 4 |
| **Basic** | 100 | 96.6 | 100 | 100 | 100 | 100 | 96.6 | 96.6 |
| **Regular** | 35.7 | 35.7 | 35 | 33.6 | 32.8 | 33.6 | 35 | 35 |
| **Extension** | 95 | 97 | 94 | 95 | 95 | 99 | 93 | 94 |
| **CFI** | 3 | 3 | 4 | 4 | 4 | 6 | 6 | 5 |
| **Total** | 52 | 52 | 51.75 | 51.5 | 51.25 | 52.5 | 51.5 | 51.5 |

### E.2. ZINC

We further evaluate $k$-harmonics with MPNNs, mirroring our extraneous BREC experiments. In Table 6 we vary $k$ and the number of message-passing layers. Effective resistance ($k = 1$) attains the best ZINC performance (both here and in Table 2) suggesting that connectivity is important in this task. Notably, a 4-layer MPNN is already competitive with attention-based models. Table 5 then shows that, while transformer variants with PEs lead in raw accuracy, augmenting a lightweight MPNN with $k$-harmonic distances recovers most of the performance at substantially lower cost with approximately $1/5^{\text{th}}$ the parameters and $1/10^{\text{th}}$ the runtime (on identical hardware). This supports the view that much of the gain comes from the $k$-harmonic PE itself, rather than the specific backbone. Baselines are drawn from (Rampasek et al., 2022; Dwivedi et al., 2023; Ying et al., 2021; Black et al., 2024b)

*Table 4.* MAE for ZINC. Results are averaged across 10 seeds.

| $k$ | **1 Layer** | **2 Layers** | **4 Layers** |
|---|---|---|---|
| $k = 1$ | $0.244 \pm 0.005$ | $0.144 \pm 0.005$ | $\mathbf{0.127 \pm 0.004}$ |
| $k = 2$ | $0.368 \pm 0.017$ | $0.188 \pm 0.006$ | $0.157 \pm 0.006$ |
| $k = 3$ | $0.401 \pm 0.008$ | $0.319 \pm 0.020$ | $0.495 \pm 0.417$ |
| $k = 4$ | $0.504 \pm 0.063$ | $0.797 \pm 0.493$ | $1.133 \pm 0.434$ |

*Table 5.* Test MAE for ZINC compared against number of parameters. The parameter to performance ratio is calculated as $(1/ \text{ test MAE}) \times$ # parameters (in millions), where higher is better.

| | Resistance MPNN | Biharmonic MPNN | Resistance Transformer | Biharmonic Transformer | Graphormer | GraphGPS | GCN-PE | GAT | Multiple Truncated PEs |
|---|---|---|---|---|---|---|---|---|---|
| Test MAE | 0.127 | 0.157 | 0.106 | 0.132 | 0.122 | **0.071** | 0.214 | 0.384 | CHANGE |
| # Parameters | **95,601** | **95,601** | 573,922 | 573,922 | 489,321 | 423,717 | 505,011 | 531,345 | CHANGE |
| Performance to Parameter Ratio | **82.36** | 66.63 | 16.44 | 13.20 | 16.75 | 33.24 | 9.25 | 4.90 | CHANGE |

### E.3. ogbg-molhiv

Lastly, we evaluate on ogbg-molhiv, a binary classification benchmark predicting whether a molecule inhibits HIV replication. This complements Table 4, where effective resistance ($k = 1$) emerged as the most suitable $k$-harmonic. In contrast, molhiv favors the biharmonic distance, indicating that node-centrality signals are comparatively more informative here than

pairwise connectivity. The main results in Table 6 show that biharmonic distance attains the best absolute performance with two message-passing layers (rather than four), consistent with its more global inductive bias. All reported improvements are statistically significant (paired Wilcoxon, $\alpha = 0.05$).

*Table 6.* % AUC for ogbg-molhiv. $k = [1, 4]$ refers to appending all $k$-harmonic distances from 1 to 4 together. Results are averaged across 10 seeds. Experiments run on MPNNs

| $k$ | **1 Layer** | **2 Layers** | **4 Layers** |
|---|---|---|---|
| No $k$-Harmonic | $74.2 \pm 1.4$ | $74.3 \pm 1.6$ | $72.5 \pm 3.5$ |
| $k = 1$ | $73.6 \pm 1.6$ | $75.5 \pm 1.3$ | $71.1 \pm 3.7$ |
| $k = 2$ | $75.7 \pm 2.1$ | $\mathbf{78.2 \pm 1.4}$ | $74.4 \pm 2.8$ |
| $k = 3$ | $74.8 \pm 1.5$ | $74.7 \pm 1.7$ | $74.6 \pm 2.4$ |
| $k = 4$ | $73.7 \pm 0.9$ | $72.6 \pm 2.1$ | $70.6 \pm 5.6$ |
| $k = [1, 4]$ | $73.7 \pm 1.3$ | $73.5 \pm 1.7$ | $73.4 \pm 1.4$ |

**Settings, Hyperparameters, and Hardware**    The settings and hyperparameters for any given experiment are contained in the configuration files that accompany our code.

All experiments were run on a single NVIDIA V100 GPU with 32GB of VRAM.

**Code**    Our code builds off of the code for the papers (Rampasek et al., 2022; Black et al., 2024b; Müller et al., 2024). Use of the code is allowable under the MIT Licensing present

https://github.com/jamesflora/understanding_truncated_positional_encodings_for_gnns

