# OpenReview forum: "Understanding Truncated Positional Encodings for Graph Neural Networks"
_ICML.cc/2026/Conference — ICML 2026 regular_

### Official Review · Reviewer_BoVg · 2026-03-10

**Soundness:** 3
**Presentation:** 3
**Significance:** 3
**Originality:** 3
**Overall Recommendation:** 4
**Confidence:** 2

**Summary:**

The authors theoretically and empirically investigate truncated positional encodings for GNNs, demonstrating that the expressive equivalence between spectral and walk-based encodings is no longer maintained under truncation.

**Compliance With Llm Reviewing Policy:**

Affirmed.

**Key Questions For Authors:**

See Weaknesses please.

**Limitations:**

The paper does not include discussion of its limitations.

**Strengths And Weaknesses:**

**Strengths**


- [S1]. The paper is well written, and I find the study of the effects of truncated positional encodings to be both interesting and novel.

- [S2]. The theoretical contributions are interesting, particularly Theorem 4.1.

- [S3]  The recommendation to mix several families of truncated PEs is interesting and consistent with the empirical results, especialyl on the ZINC dataset.

**Weaknesses**

- [W1] The empirical section is interesting, but its scope remains somewhat limited relative to the theoritical claims of the paper. In particular, the recommendation to combine truncated PEs from different families is mainly supported by the results on ZINC but in my opinion It would also be interesting to show that the benefit of combining truncated PEs is robust across different backbones. Broadening the experimental evaluation to a wider range of benchmarks would strengthen the conclusions of the work.

---

> ### Author Rebuttal · Authors · 2026-03-31
>
> **[W1] Scope of Experiments**
>
> We thank the reviewer for this comment and agree that broader evaluation across more benchmarks and backbones would strengthen the empirical case. However, we would also like to stress that the scope was deliberate: the main novelty of the paper is theoretical, in that we initiate the study of truncated positional encodings and their combinations, and experiments are intended primarily to validate and illustrate that theory. In that sense, BREC and ZINC were chosen to exemplify two complementary aspects of the story: theoretical expressivity and practical downstream utility (we further note the experiments on ogb-molhiv in the appendix). We agree that robustness across more architectures would be valuable future work, but we believe that the current experiments validate our central theoretical findings: truncated PE families can encode complementary structural information and combining them can be beneficial in practice.

---

> > ### Author Rebuttal · Reviewer_BoVg · 2026-04-03
> >
> > I thank the authors for their responses.
> >
> > After reading the other reviews, and given my low confidence, I would like to maintain my score.
> >
> > I wish the authors the best of luck for the future.

---

### Official Review · Reviewer_4vxx · 2026-03-10

**Soundness:** 2
**Presentation:** 3
**Significance:** 1
**Originality:** 4
**Overall Recommendation:** 4
**Confidence:** 4

**Summary:**

This paper addresses the expressivity of message-passing graph neural networks endowed with spectral and walk-based positional encodings. In particular, it focuses on the truncated versions of these encodings (e.g., using the first or last k eigenvectors in spectral ones, or only random walks up to length k in RWPE), which are the versions used in practice. The paper also studies harmonic distances (an extension of effective resistance) as positional encodings, and proves positive results on the expressive power of these encodings. The theoretical results are complemented by empirical results on the BREC dataset and on the ZINC dataset, showing that ensembles of positional encodings are in general better than single ones.

**Compliance With Llm Reviewing Policy:**

Affirmed.

**Final Justification:**

Despite some remaining issues I raised my score, in good faith that the authors will include the two important discussions mentioned in my reply.

**Key Questions For Authors:**

See weaknesses.

**Limitations:**

Yes (Appendix B.2)

**Strengths And Weaknesses:**

Strengths:
- the paper addresses a gap in the literature between the expressivity analysis of PEs and the PEs that are used in practice.
- the empirical results (as well as the theoretical ones) on ZINC for the harmonic distances are very promising, and could inspire new research in the development of novel effective PEs.
- the paper is quite well-written and clear.
- the contribution is original and the formal statements in general seem to be correct.

Weaknesses:
- In general, I’m really unsure about the usefulness of these expressivity results.
    - First, most real-world graphs are distinguishable by 1-WL [1, 2]. Therefore, PEs are not needed to increase expressivity. This should be discussed in the paper. I think that the usefulness of PEs more likely comes from the fact that PEs correlate well with the labels that are commonly of interest.
    - Secondly, the WL-like tests defined here seem to be at least partially detached from reality. For example, if a GNN (say, a GIN) uses the PEs as additional node or edge features, as common, then it would be automatically at least as strong as WL. This somewhat contradicts the statement that adding PEs yields expressive power incomparable with WL (as implied by the result of Theorem 4.1 and the subsequent discussion), which focuses on an ad-hoc architecture.
    - While Graphormer-GD is equivalent to the ψ-WL test, commonly used graph transformers, such as GraphGPS, also use some message-passing mechanism, and I suspect that this would put them back into the WL hierarchy.
- The experimental results are not really convincing. While BREC is interesting as a proof-of-concept, it cannot be considered one of the main datasets on which to benchmark methods. Indeed, the types of graphs seen in BREC have very little practical relevance. The results on the only real-world dataset in the main paper, ZINC, as I said in the “strengths”, are more convincing. However, a more thorough evaluation on additional datasets would strengthen the contribution.
- Interestingly, the results on molhiv in the appendix show predictive performance that is much more unstable compared to the results on ZINC, and is further away from SOTA results [3]. This is not discussed properly in the paper.

.

- [1] Zopf. 1-WL Expressiveness Is (Almost) All You Need. IJCNN 2022
- [2] Pellizzoni et al. Graph Neural Networks Can (Often) Count Substructures. ICLR 2025.
- [3] Luo et al. Can Classic GNNs Be Strong Baselines for Graph-level Tasks? Simple Architectures Meet Excellence. ICML 2025.

---

> ### Author Rebuttal · Authors · 2026-03-31
>
> **[W1] Most real-world graphs are distinguishable by 1-WL**
>
> We agree that in many real-world benchmarks, most graphs are already distinguishable by 1-WL. However, we do not believe this makes expressivity results irrelevant. Our use of expressivity is not meant to suggest that downstream tasks consist of deciding graph isomorphism; rather, it serves as a principled way to characterize which structural signals a positional encoding may provide to the model. In practice, these signals need not determine whether two graphs are distinguishable, but may instead correspond to important properties such as centrality, or long range dependencies, which can be highly predictive of labels. For example, Theorem 4.1 shows that k-EP-WL is not stronger than 1-WL. This implies that networks using k-EP-WL cannot detect which pairs of nodes are connected by an edge.
>
> In this sense, expressivity is a proxy for structural sensitivity. The observation that useful PEs correlate with task-relevant labels is compatible with our perspective: the reason a PE may correlate with those labels is precisely because it captures structural information that the base architecture would otherwise struggle to access.
>
> We agree that this distinction could be made more explicit in the paper.
>
> **[W2, W3] Definition of WL-like tests, GraphGPS**
>
> We would like to politely object to the claim that our results are “detached from reality”,  as there is a large body of work studying graph transformers that do not incorporate any form of message passing. We acknowledge that GraphGPS or GIN are common architectures and that it is true that using a GIN or GPS would make an the architecture at least as powerful as WL, However, there are many common graph transformer architectures that do not use message passing ([1], [2]). In particular, the Graphormer paper has twice as many citations as GraphGPS, so it is far from the case that all, or even most, graph transformers use message-passing. We use a GD-Graphormer type architecture [2], which is a common transformer architecture for relative positional encoding and not an “ad-hoc architecture”. Moreover, a recent ICML 2025 [3] paper shows that the GD-Graphormer architecture is equivalent to a simpler transformer that only uses relative positional encoding inside the softmax. Therefore, our results apply to most transformers that use relative positional encodings, including architecture like Graphormer. (We will add a discussion of our specific architecture to future versions of our paper.)
>
> That said, our results actually provide an argument in favor of GraphGPS. As Theorem 4.1 shows that truncated eigenspace PEs alone are not stronger than 1-WL, this implies that message-passing is a necessary architectural component for GraphGPS to be more powerful than 1-WL, and removing the message-passing layers from GraphGPS would lower its expressive power.
>
> Finally, we do consider the power of using PEs with message-passing architecture like GINs. In Section 4.4, we consider the sparse-WL test, which is the name we give to the WL test that only uses the value of RPEs on edges. The sparse-WL test characterizes GINs with edge features, and we prove some theorems for MPNNs that use the k-harmonic distances as edge features.
>
> **[W4] Experimental Datasets and Generality**
>
> We appreciate this feedback. Our intention in including BREC was to use it as a controlled, generalized measure of expressivity beyond what proofs can provide. We agree, however, that broader evaluation would strengthen the paper, and refer the reviewer to our response to Reviewer 1 for discussion of scope.
>
> Regarding ogbg-molhiv, we agree that the results are further from SotA than those on ZINC. However, our goal was not to claim competitive performance, but rather to study the behavior of the k-harmonic family itself. In particular, the model used here is a vanilla MPNN augmented only with the k-harmonic distances, so it is not surprising that it remains below more sophisticated transformer-style architectures.
>
> The main takeaway we intended is therefore not absolute performance, but the fact that the preferred value of k differs from what we observe on ZINC. This supports our broader claim that different k-harmonics capture different structural information, and that the usefulness of a given truncated PE is task-dependent. We will state this more clearly.
>
>
> [1] Ying, C., Cai, T., Luo, S., Zheng, S., Ke, G., He, D., ... & Liu, T. Y. (2021). Do transformers really perform badly for graph representation?. Advances in neural information processing systems, 34, 28877-28888.
>
> [2] Zhang, B., Luo, S., Wang, L., & He, D. Rethinking the Expressive Power of GNNs via Graph Biconnectivity. In The Eleventh International Conference on Learning Representations.
>
> [3] Stoll, T., Müller, L., & Morris, C. Generalizable Insights for Graph Transformers in Theory and Practice. In The Thirty-ninth Annual Conference on Neural Information Processing Systems.

---

> > ### Author Rebuttal · Reviewer_4vxx · 2026-04-02
> >
> > [W4] I still find the experimental evaluation weak, but for a theory-driven paper I understand that it is secondary.
> >
> >
> > [W2, W3] Don't get me wrong, I agree that studying the expressivity of Graphormer is interesting, I just found that your results can be applied only in a very narrow set of architectures. Expanding to combinations of positional encodings and message passing would be interesting. Also usually in a Graphormer one would also include edge connectivity information, which should recover a message-passing-like expressivity (see Fact 1 in [1]). That's why I find your results a bit narrow. I believe that an honest discussion of these limitations in the papers would make it more valuable for the reader, and a great starting point for future research.
> >
> > [W1] I agree with you that expressivity is not irrelevant. Nonetheless, I find that a discussion on the distinguishability of real-world data and thus the use of expressivity as a proxy of structural sensitivity would be really valuable to the reader to situate your results in the broader picture of graph learning.
> >
> >
> > Despite some remaining issues I will raise my score, in good faith that you will include the two important discussions mentioned above.

---

> > > ### Author Response · Authors · 2026-04-06
> > >
> > > Thank you for the thoughtful follow-up and for the constructive discussion throughout the review process. We sincerely appreciate your willingness to revise your score.
> > >
> > > We agree that the points you have highlighted are important for properly situating the paper. We will incorporate discussions of both into the final version of the paper so that the scope, limitations, and contribution are as clear as possible.

---

### Official Review · Reviewer_U1GB · 2026-03-11

**Soundness:** 3
**Presentation:** 3
**Significance:** 3
**Originality:** 3
**Overall Recommendation:** 5
**Confidence:** 3

**Summary:**

the paper shows that under truncation, several families of PEs are fundamentally different in expressive power compared with full PE. also, experiments show that a mix of truncated PEs is preferable to any single family on real-world datasets.

**Compliance With Llm Reviewing Policy:**

Affirmed.

**Final Justification:**

i've read the author rebuttal, will keep my rating as accept.

**Key Questions For Authors:**

how does the behavior of k-harmonic PEs on very sparse or very dense social networks look like?

**Strengths And Weaknesses:**

strength:
1. the paper is well written and easy to follow
2. Experiments on BREC and ZINC show that combining truncated encodings from different families improves performance.

---

> ### Author Rebuttal · Authors · 2026-03-31
>
> **[Q1] k-harmonic PEs on Social Networks**
>
> We thank the reviewer for this interesting question. At present, we do not have a definitive empirical characterization of how k-harmonic PEs behave specifically on sparse versus dense social networks, but our results suggest several plausible trends.
>
> First, different k-harmonics capture different types of structural information. For example, effective resistance measures pairwise connectivity, or how well two nodes are linked through the surrounding graph. This could be used to measure how “close” two friends are in a social network; the closer two people are in effective resistance, the more mutual connections they have, which matches typical notions of closeness in social relationships. Alternatively, biharmonic distance is high on edges between different clusters [1]. In the context of social networks, biharmonic distance could be used to detect relationships between members of different communities. For k > 2, the intuitive explanation is less developed, though our theoretical results demonstrate that these k also are powerful features for different graph tasks.
>
> With respect to graph density, one challenge is in the amount of computation required to process and train graphs on the order of magnitude common in social networks. Dense PEs on this order of magnitude quickly become infeasible at this scale. This is precisely one of our motivations for focusing on truncated positional encodings, including the k-harmonics: they offer a way to retain useful information in the preprocessing step at a reduced cost (see: Appendix D.8) and reduce training time (please see our response to Reviewer #1, and Appendix D.1 for the sparse variant of this test).
>
> Regardless, we do not yet claim a concrete characterization for social networks, but we view the k-harmonic distances as a promising tool for future studies on the subject.
>
> 1. Black, M., Lin, L., Wong, W.-K., and Nayyeri, A. Biharmonic distance of graphs and its higher-order variants: Theoretical properties with applications to centrality and clustering.

---

> > ### Author Rebuttal · Reviewer_U1GB · 2026-04-04
> >
> > i've read the author rebuttal, will keep my rating as accept.

---

### Official Review · Reviewer_FxSk · 2026-03-13

**Soundness:** 3
**Presentation:** 2
**Significance:** 3
**Originality:** 3
**Overall Recommendation:** 4
**Confidence:** 2

**Summary:**

The paper analyzes three families of positional encodings: spectral encodings (eigenspace projections), walk-based encodings (powers of the adjacency matrix), and resistance-based encodings including k-harmonic distances. The authors show that while these encodings have equivalent expressive power in their full form, truncating them leads to significant differences in expressive power.

The paper provides several theoretical results comparing the expressiveness of these truncated encodings relative to the WL hierarchy, and shows experimentally that combining positional encodings from different families tends to outperform using a single truncated encoding.

**Compliance With Llm Reviewing Policy:**

Affirmed.

**Key Questions For Authors:**

1. Many theoretical results rely on carefully constructed graph examples. How often do these pathological cases appear in real-world datasets used for graph learning?

2. Have the authors tested the proposed PE combinations on additional datasets beyond BREC and ZINC to verify the generality of the findings?

3. Some positional encodings (like spectral methods) require expensive preprocessing. How does the proposed mixture approach scale to very large graphs?

4. The paper suggests mixing positional encodings from different families. Are there guidelines for choosing which encodings to combine in practice?

**Limitations:**

The paper does not discuss limitations or potential societal impacts of the work.

The authors could discuss the computational cost of computing positional encodings, particularly spectral encodings that require eigenvalue decompositions or Laplacian pseudoinverses. Although truncated encodings mitigate this issue, they may still be expensive for very large graphs.

**Strengths And Weaknesses:**

1. Soundness:

The paper presents several theoretical results regarding the expressive power of truncated positional encodings and their relationship to WL-type tests. The arguments appear technically motivated and are supported by formal theorems and proofs.

However, some aspects have concerns. First, the results largely focus on worst-case distinguishability of graph pairs, which may not necessarily translate into practical improvements in learning tasks. While theoretical expressivity is important, the connection between these results and real-world performance is somewhat indirect.

Second, although the paper includes experiments, they are relatively limited and primarily focus on benchmark datasets such as BREC and ZINC. More extensive empirical evaluation across different graph learning tasks would strengthen the claims regarding practical usefulness.

2. Presentation:

The paper is generally well structured and logically organized. The motivation behind studying truncated positional encodings is clearly shown.
However, the paper is quite dense in several theoretical sections. Readers who are not already familiar with these concepts may find parts of the exposition difficult to follow. Providing more intuitive explanations alongside the formal theorems would improve accessibility.

3. Significance:

Understanding the expressive power of positional encodings is an important topic for graph neural networks and graph transformers. Since truncated encodings are widely used in practice due to computational constraints, analyzing their theoretical properties is a relevant and meaningful research direction.
However, the practical impact may depend on whether these theoretical insights translate consistently into improvements across a wider range of tasks.

4. Originality:

The analysis comparing spectral, walk-based, and resistance-based encodings under truncation is interesting and provides new insights into how these representations differ in expressive power. The introduction of k-harmonic distances as a bridge between different encoding families is also an important aspect of the work. Overall, while the paper builds on existing theory about WL tests and spectral graph representations, it contributes new perspectives on how these tools behave under practical constraints.

---

> ### Author Rebuttal · Authors · 2026-03-31
>
> **[Q1] Theoretical results rely on carefully constructed counterexamples**
>
> We thank the reviewer for this question. First, we agree that many expressivity results rely on carefully constructed graph examples. Their purpose, however, is to isolate structural phenomena that a given PE can or cannot capture, not to necessarily claim that these graphs are present in real-world datasets. For example, Theorem 4.1 shows that k-EP-WL is not stronger than 1-WL. This implies that networks using k-EP-WL cannot detect which pairs of nodes are connected by an edge. Additionally, our experiments on the BREC dataset, which contains many different pairs of graphs between 1-WL and 3-WL indistinguishable, shows that there are pairs of graphs beyond our specific counterexamples that one PE can distinguish while others cannot.
>
> **[Q2] Additional datasets to verify generality**
>
> We have not yet evaluated the proposed PE combinations on datasets beyond BREC and ZINC, though we have a short experiment on ogb-molhiv in Appendix E.3. This was a deliberate choice in scope. The primary contribution of the paper is to initiate the study of these truncated positional encodings from a theoretical perspective, and our empirical goal was therefore to provide enough evidence to validate and contextualize the theory. The experiments were chosen along two complementary axes: BREC provides a controlled benchmark for fine-grained expressivity beyond 1-WL, while ZINC serves as a standard real-world regression task for assessing practical utility. Together, these datasets test the proposed combinations in two settings that are different in objective and structure.
>
> **[Q3] Preprocessing and scaling Truncated PEs**
>
> We agree that this is an important point on the scalability of truncated PEs. In the paper, we partially address this in Appendix D.8 by an approximation algorithm and runtime analysis of computing the k-harmonic distances. For the other PEs (random walks and eigenspace projections), the runtime for computing these is known (matrix multiplication for Walk PE or matrix decomposition for eigenspaces. While there are some potential optimizations for these PEs too (sparse matrix multiplication for Walks, power method/alternative algorithms for eigenvectors, etc), they both have naive runtimes of O(n^3).
>
> That said, we emphasize that the dominant cost for transformers is often not the preprocessing itself, but rather the training time once dense PEs are incorporated into the model. As graph size and PE size grow, the pipeline will generally be bottlenecked by transformer training time long before it is bottlenecked by the preprocessing step, as the runtime of each attention computation is proportional to the size of the PE, which is O(n^3) for the full case but O(kn^2) for the truncated case. From this perspective, truncation is valuable not only because it reduces the one-time preprocessing cost, but more importantly because it reduces the training cost.
>
> As such, our work seeks to evaluate the accuracy-efficiency tradeoff to be made when looking at these mixtures of PEs versus a single large PE: a small number of complementary signals can be more effective than scaling a single PE in isolation.
>
> **[Q4] Guidelines on Mixing PEs**
>
> We expect the best PE combination to be both dataset and task-dependent. One of the central messages of the paper is precisely that different positional encodings capture different kinds of structural information, so there is no reason to expect a single combination of truncated PEs to be optimal. Our k-harmonic results provide a concrete example: different values of k encode different spectral information, and empirically we observe that different datasets prefer different choices (e.g., ZINC favors k=1, while ogb-molhiv favors k=2). This suggests more broadly that the most useful structural signal depends on the downstream task.
>
> At the same time, we would emphasize that the novelty of our work is not in prescribing a definitive mixture, but in initiating the study of combining multiple truncated PE families in the first place. Our practical takeaway is therefore intentionally modest: rather than searching for one universally best PE, we recommend combining a small number of complementary truncated encodings.
>
> Determining which combinations are optimal for specific datasets, and understanding more precisely why those preferences arise per task, is an important direction for future work.

---

> > ### Author Rebuttal · Reviewer_FxSk · 2026-04-04
> >
> > Thank you for your responses.
> >
> > While the rebuttal improves the clarity of the work and addresses several of my concerns, some important issues remain insufficiently clarified. As such, I will keep my original score unchanged.

---

### Decision · Program_Chairs · 2026-04-30

**Decision:**

Accept (regular)

**Comment:**

Reviewers agreed that the paper addresses an important and timely question: how truncation changes the expressive power of practical positional encodings for graph models. They found the theoretical results novel and technically strong, while noting that the empirical section is limited in breadth and that the discussion of practical implications and architectural scope could be expanded. The rebuttal satisfactorily addressed most questions, and no reviewer raised a fundamental correctness issue after discussion. Overall, I recommend Accept because the paper delivers a solid theoretical contribution with clear insight for future design of positional encodings, even if the empirical section could be broader.